# Meridional energy transport extremes and the general circulation of Northern Hemisphere mid-latitudes: dominant weather regimes and preferred zonal wavenumbers

Valerio Lembo[1], Federico Fabiano[1], Vera Melinda Galfi[2], Rune Graversen[3,4], Valerio Lucarini[5], and Gabriele Messori[2,6]

[1]Consiglio Nazionale delle Ricerche, Istituto di Scienze dell'Atmosfera e del Clima (CNR-ISAC), Bologna, Italy
[2]Department of Earth Sciences and Centre of Natural Hazards and Disaster Science (CNDS), Uppsala University, Uppsala, Sweden
[3]Department of Physics and Technology, UiT – The Arctic University of Norway, Tromsø, Norway
[4]Norwegian Meteorological Institute, Tromsø, Norway
[5]Department of Mathematics and Statistics and Centre for the Mathematics of Planet Earth, University of Reading, Reading, UK
[6]Department of Meteorology and Bolin Centre for Climate Research, Stockholm University, Stockholm, Sweden

**Correspondence:** Valerio Lembo (v.lembo@isac.cnr.it)

**Abstract.** The extratropical meridional energy transport in the atmosphere is fundamentally intermittent in nature, having extremes large enough to affect the net seasonal transport. Here, we investigate how these extreme transports are associated with the dynamics of the atmosphere at multiple spatial scales, from planetary to synoptic. We use the ERA5 reanalysis data to perform a wavenumber decomposition of meridional energy transport in the Northern Hemisphere mid-latitudes during winter and summer. We then relate extreme transport events to atmospheric circulation anomalies and dominant weather regimes, identified by clustering 500hPa geopotential height fields. In general, planetary-scale waves determine the strength and meridional position of the synoptic-scale baroclinic activity with their phase and amplitude, but important differences emerge between seasons. During winter, large wavenumbers ($k = 2 - 3$) are key drivers of the meridional energy transport extremes, and planetary and synoptic-scale transport extremes virtually never co-occur. In summer, extremes are associated with higher wavenumbers ($k = 4 - 6$), identified as synoptic-scale motions. We link these waves and the transport extremes to recent results on exceptionally strong and persistent co-occurring summertime heat waves across the Northern Hemisphere mid-latitudes. We show that the weather regime structures associated with these heat wave events are typical for extremely large poleward energy transport events.

## 1 Introduction

The latitudinal gradient in incoming net solar radiation is the main trigger of atmospheric and oceanic dynamics (Peixoto and Oort, 1992). Indeed, the resulting energy imbalance induces an atmospheric circulation that causes divergence of heat at low latitudes and convergence at high latitudes. In the low latitudes, a key role is played by the zonally averaged circulation,

dominated by the Hadley circulation, and the energy transport embedded in the mass streamfunction (Holton and Hakim, 2012). In the mid-latitudes, eddies are the most significant contributors to the overall transport.

Meridional energy transports through eddies are increasingly recognized as communicating the climate change signal towards the high latitudes, especially when taking into account the contribution by moisture (Hwang et al., 2011; Skific and Francis, 2013; Baggett and Lee, 2015; Rydsaa et al., 2021). Conventionally, the eddy contribution is partitioned into a transient and a stationary component, with the former identified as the instantaneous deviation from long-term climatology, and the latter as the climatological departure from the zonal mean transport (e.g. Starr and White (1954); Lorenz (1967)). Energy transports

associated with transient eddies have been found to exhibit a large variability, with a number of sporadic extreme events exceeding the mean values by a few orders of magnitude (Swanson and Pierrehumbert, 1997; Messori and Czaja, 2013, 2015). This emphasizes the non-linear nature of baroclinic activity and its interactions at different wave-scales and with the zonal-mean circulation (cfr. Kaspi and Schneider (2013); Messori and Czaja (2014); Novak et al. (2015)).

As an alternative to the usual transient-stationary eddies partitioning, a Fourier decomposition of eddy modes, isolating the

contribution of different zonal wavenumbers to the overall transport has been proposed (Graversen and Burtu, 2016). Such zonal wavenumber decomposition relies on zonally integrated fluxes, meaning that spatially localised information is lost, and the contribution of different wavenumbers is provided in terms of the transport they effect across a given latitudinal circle. Despite this, it is still possible to link the results of the zonal wavenumber decomposition to the mid-latitude atmospheric circulation. For example, Graversen and Burtu (2016) found that latent energy convergence through planetary scale waves is

the most important contribution to the Arctic amplification, in agreement with Baggett and Lee (2015); Boisvert and Stroeve (2015). Heiskanen et al. (2020) have also shown that planetary-scale waves have significantly increased their contribution in the last decades (cfr. Rydsaa et al. (2021)). These findings are placed in the context of ongoing debate on whether the decreasing meridional temperature gradient determined by Arctic amplification is setting more favorable conditions for the propagation and growth of planetary-scale waves (cfr. Barnes and Polvani (2013); Fabiano et al. (2021); White et al. (2022); Moon et al.

40   (2022)).

By separating synoptic and planetary scales as having wavenumbers $k > 5$ or $k = 1 - 5$, respectively, it has been recently suggested that synoptic-scale eddies always allow for baroclinic conversion, and thus systematically contribute to a poleward transport, while planetary-scale eddies can occasionally oppose the total transport resulting from the sum of all contributions (Lembo et al., 2019). A pronounced seasonal variability was also noticed, with planetary scales undergoing a much larger de-

crease from winter to summer than synoptic scales. Separating synoptic and planetary-scale waves by wavenumber is inevitably arbitrary (Heiskanen et al., 2020; Stoll and Graversen, 2022); nonetheless, the results from (Lembo et al., 2019) point towards the importance of modes of low-frequency/large-scale variability in modulating the energy transport by eddies (Messori and Czaja, 2015; Messori et al., 2017) and the emergence of preferred weather regimes in the occurrence of intermittent meridional energy transport extreme events (cfr. Nie et al. (2008); Ruggieri et al. (2020)). In a quasi-geostrophic context, the conversion

of available potential energy into kinetic energy (Lorenz, 1955) supports interpreting the energy transfers in terms of modes of atmospheric variability. This then enables to link meridional energy transport variability to regional climate variability and

extremes, as particularly evident for climate extremes associated with specific circulation features, such as summertime blocks leading to heatwaves (e.g. the 2010 summer event over Russia; Dole et al. (2011); Galfi and Lucarini (2021)).

Here, we focus on meridional energy transports in the mid-latitudes, where the baroclinic activity is strongest. We decompose the transport using the above-described zonal wavenumber decomposition approach of Graversen and Burtu (2016). Given the intermittent and sporadic nature of eddy-driven meridional energy transports described in Woods et al. (2013); Messori and Czaja (2013) and Messori and Czaja (2015), we focus on meridional energy transport extremes and analyse the associated circulation anomalies and weather regimes. Despite the known role of synoptic weather systems (e.g. Liu and Barnes (2015)) and storm tracks (e.g. Dufour et al. (2016)) for the transfer of excess heat into the Arctic, our understanding of how these and other features of the mid-latitude atmospheric circulation may favour extreme transports is far from complete. We select the extremes with a rigorous methodology based on Extreme Value Theory (EVT) (Gnedenko, 1943). The extreme events are then associated with preferred patterns of the general circulation through k-means clustering of 500 hPa geopotential anomalies, identifying a set of weather regimes. The frequency of occurrence of such regimes is used to interpret the atmospheric circulation features associated with the occurrence of extremes in zonally integrated meridional energy transport. A composite mean analysis is also provided, highlighting some persistent patterns in specific regions of the NH (Northern Hemisphere) midlatitudes.

The paper is organised as follows: in Sect. 2 we describe the data, the methodology for extreme events selection and for the clustering of weather regimes. In Sect. 3a distributions of meridional energy transports, their wavenumber decomposition and their extremes as a function of latitude are described, while in Sect. 3b clustering of geopotential heights is related to composites of extreme events and dominant wavenumbers. In Sect. 4 an interpretation of these results is given, in light of composite mean anomaly patterns of geopotential height fields associated with extreme transports, while Sect. 5 summarises the findings and outlines limitations and future perspectives.

## 2 Methodology

### 2.1 Data

Data for meridional energy transport and atmospheric circulation analyses are obtained from the ERA5 Reanalysis (Hersbach et al., 2020), retrieved from the Copernicus Data Storage (CDS). The data used covers December-January-February (DJF) and June-July-August(JJA) from 1979 to 2012, with a 6-hourly temporal resolution, a horizontal spatial resolution of $1° \times 1°$ and 137 vertical levels. We deem the 6-hourly sufficient to encompass the synoptic and larger modes of variability, as evidenced in recent analyses (Lembo et al., 2019). The choice of seasons is aimed at avoiding conflating regimes that are representative of the warm and cold seasons, as may happen in spring and autumn. The analysis is focused on the mid-latitudinal Northern Hemisphere band, i.e. latitudes in the $30°$ N–$60°$ N range. For the computation of moist static energy, specific humidity (q), air temperature (T), and geopotential height (z) are considered. For the retrieval of the weather regimes and the $k$-means clustering, daily geopotential height at 500 hPa (z500) is used.

## 2.2 Methods

### 2.2.1 Meridional energy transports and wavenumber decomposition

The instantaneous meridional energy transport (or, more correctly, the enthalpy transport, cfr. Ambaum (2010); Lucarini and Ragone (2011)) is obtained as the scalar product of the meridional velocity $v$ and the total energy $E$, including the kinetic energy and the moist static energy $H$, defined as:

$$H = L_v q + c_p T + gz \tag{1}$$

where $q$ is the specific humidity, $T$ is the air temperature, $z$ is the geopotential height, $L_v$, $c_p$ and $g$ are the latent heat of vaporization, the specific heat at constant pressure, and the gravity acceleration, respectively. The transport of $E$ across a given latitude is computed with the vertical and zonal integration:

$$\oint \int_{p_s}^{0} vE \frac{dp}{g} dx = \oint \int_{p_s}^{0} v(H + \frac{1}{2}\mathbf{V}^2) \frac{dp}{g} dx \tag{2}$$

where $p_s$ is the surface pressure and $\mathbf{V}$ is the horizontal velocity vector. The sign convention is that transport is positive when northward directed, and negative when southward directed. The wavenumber decomposition is extensively discussed in Graversen and Burtu (2016), and in Lembo et al. (2019). Here, we recall that the meridional energy transport as a function of zonal wavenumber $k$ can be written as:

$$\hat{\mathcal{F}}_0(\phi) = D \int_{p_s}^{0} \frac{1}{4} a_0^v a_0^E \frac{dp}{g} \qquad\qquad k = 0 \tag{3}$$

$$\hat{\mathcal{F}}_k(\phi) = D \int_{p_s}^{0} \frac{1}{2} (a_k^v a_k^E + b_k^v b_k^E) \frac{dp}{g} \qquad\qquad k = 1, \dots, N \tag{4}$$

where $a$ and $b$ are the Fourier coefficients defined as:

$$a_k^\Psi(t, \phi) = \frac{2}{D} \int \Psi(t, \phi, \lambda) \cos\left(\frac{k2\pi\lambda}{d}\right) d\lambda \tag{5}$$

$$b_k^\Psi(t, \phi) = \frac{2}{D} \int \Psi(t, \phi, \lambda) \sin\left(\frac{k2\pi\lambda}{d}\right) d\lambda \tag{6}$$

where $\Psi$ is either $v$ or $E$, $D = 2\pi R \cos(\phi)$ with $R$ being the Earth's radius, $k$ being the zonal wavenumber, and $t$, $\phi$, $\lambda$ the time, latitude and longitude, respectively, and $d = 360^\circ$. The simple form in Eq. 4 is made possible by the fact that the k-members of the Fourier decomposition are harmonics in a cylindrical symmetry. Parseval's theorem then ensures that the cross-terms in the multiplication of the two Fourier series cancel each other when integrated over a latitude circle (Tolstov, 2012).

The highest zonal wavenumber $N$ is set to 20, given that the spatial and temporal resolution of the ERA5 outputs poses a constraint to the modes of variability that can be resolved, and since it has been shown that the transport associated with wave number 0 to 20 constitute almost entirely the total transport (Graversen and Burtu, 2016). For sake of conciseness, we group

wavenumbers into: $k=0$, reflecting zonal mean transports, $k=1-5$ for planetary-scale transports, $k=6-10$ for synoptic-scale transports and $k=11-20$ as mesoscale and lower spatial scale transports. The choice of thresholds for separating main eddy modes is somewhat arbitrary. Here, we follow the choice made in Graversen and Burtu (2016) and subsequently in Lembo et al. (2019) to ensure easy inter-comparability of results. Although there are also arguments supporting other threshold choices (cfr. Heiskanen et al. (2020); Stoll and Graversen (2022); see also Appendix C), earlier studies of different impact of planetary

versus synoptic-scale waves have revealed that decomposition of the transport into these wave types is not dependent on the choice of the threshold for wave grouping (Heiskanen et al., 2020; Stoll and Graversen, 2022).

The meridional circulation includes a zonal mean mass transport in opposite direction at different altitudes. However, at small temporal scales a vertical mean mass transport across latitudes may be encountered, which may transport a large amount of energy. This latter component of the enthalpy transport has to be subtracted from wave zero transport, in order to obtain the

energy transport by the mean meridional circulation (Liang et al., 2017), which is done following Lembo et al. (2019). The instantaneous zonal mean and vertical mean meridional-circulation energy transport component is thus written as:

$$\{[v]^{\dagger}[E]^{\dagger}\} = \{[v][E]\} - \{[v]\}\{[E]\} \tag{7}$$

with $\{\ \}$ denoting the vertical mean, $[\ ]$ the zonal mean and $\dagger$ the departure from the vertical mean. Note that the wave zero energy transport component is given by the first term on the r.h.s. of Eq. 7.

**2.2.2 Selection of the extreme events**

We adopt in this work a rigorous methodology for the detection of meridional energy transport extremes based on the "peak over threshold" approach of Extreme Value Theory (EVT) (Balkema and Haan, 1974; Pickands, 1975). The main advantage of this methodology is that the properties of the data themselves determine the threshold used to define extreme events instead of arbitrary threshold choices. Furthermore, the probability distribution of extreme values is given by EVT, thus no subjective

decisions are needed considering the shape of this distribution. An overview of this approach is hereby drawn, based on Coles et al. (2001), and of the convergence algorithm also presented in Gálfi et al. (2017). Full theoretical and technical details can be found therein.

We first consider a series of independent identically distributed random variables $X_1, X_2, ...$ with common probability distribution $F(X)$, $X$ denoting an arbitrary term in the series $X_i$. Under appropriate conditions (discussed generally below, for a

comprehensive and rigorous description see Leadbetter (1974); Leadbetter et al. (1989); Coles et al. (2001)), the threshold exceedences $y$ of a large enough threshold $u$, $y = X - u$ with $X > u$, are asymptotically distributed according to the Generalized Pareto Distribution (GPD) family as:

$$H(y) = 1 - \left(1 + \frac{\xi y}{\sigma}\right)^{\frac{1}{\xi}} \qquad \text{for} \qquad \xi \neq 0 \tag{8}$$

$$H(y) = 1 - \exp^{-\frac{y}{\sigma}} \qquad \text{for} \qquad \xi = 0 \tag{9}$$

with $1 + \xi y / \sigma > 0$ for $\xi \neq 0$, $y > 0$, and $\sigma > 0$. The shape parameter $\xi$ determines the shape of the distribution and the tail behavior. If $\xi = 0$ the tail of the distribution decays exponentially, if $\xi > 0$ the tail decays polinomially, while, if $\xi < 0$, the distribution is bounded.

In order to apply EVT to meridional energy transport data, one has to make sure that the conditions of independence and homogeneity are fulfilled, and thus the data can be modelled based on a stationary stochastic process (Leadbetter, 1974). While both conditions are, strictly speaking, unrealistic in case of geophysical data, the problem can be approached from a practical perspective. First, we detect linear long-term trends for the 1979-2012 period, and we test their significance according to a Mann-Kendall test at the 95% significance level (Mann, 1945; Forthofer and Lehnen, 1981), then we remove trends only at those latitudes where they are found to be significant, i.e., for DJF, north of 37.5° N, for JJA in the 40° N–47.5° N and 54.5° N–58.5° N latitudinal bands. By analysing summer and winter separately, we eliminate another source of dependence and heterogeneity due to the seasonal cycle. Furthermore, we verify for every latitude the decay of the auto-correlation function. Despite the fact that several consecutive values in our times series are correlated, the chaotic nature of atmospheric motions allows that two values being far away in time from each other are nearly independent, meaning that the auto-correlation function decays to 0 at a finite time lag. Looking at the auto-correlation functions, we find that JJA (DJF) daily data require the removal of the climatological JJA (DJF) season at every latitude (south of 45° N). Given that the non-homogeneous detrending and deseasonalization may in principle introduce discontinuities, we compared results obtained with the described method, to those obtained when the two corrections were applied at all latitudes (not shown). The small discontinuity found at latitude 45° N in the meridional section of threshold values (Figure 1b) was indeed removed, when the corrections were applied at all latitudes, but the difference is minimal. Finally, the GPD parameters were computed. Even in the absence of long-range correlations, however, one may encounter clusters of data exceeding the threshold consecutively in time. Those are clearly correlated and would lead to biased GPD parameters[1] as a result of a slower convergence to the limiting distribution (Coles et al., 2001). To avoid this, we decluster, where necessary, the time series before estimating the parameters by using the "intervals" declustering method (Ferro and Segers, 2003) based on the "extremal index" defined in Gálfi et al. (2017).

We then test the applicability of the theory by looking at the convergence of the selected extremes to the limiting GPD distribution. This is done by plotting the estimated shape parameter $\xi(u)$ as a function of an increasing threshold $u$ (cfr. Figure 1). By visual inspection, for the right (left) tail of the PDF, the lowest (highest) threshold $u^*$ at which the shape parameter becomes stable, i.e. does not change with further threshold increase (decrease), provides the optimal parameter estimate. This estimate has a lower uncertainty than the ones for $u > u^*$ ($u < u^*$), and, at the same time, the distribution of the selected extremes is equivalent to the limiting GPD, in the limits of estimation uncertainty given by the confidence intervals. In order to achieve a latitude-independent definition of extremes, we express the threshold in terms of fraction of data points above (below) the threshold and the total number of data points. For each tail of the distribution, the same threshold is provided at all latitudes, given as the highest fraction of data points for which the convergence is achieved at all latitudes.

This threshold selection procedure applies in case of extremes located in the right tail of the transport's probability density function (PDF), hereafter referred to as "poleward" extreme energy transports. Similarly, we refer to extremes located in the

---

[1]The asymptotic GPD shape parameter is not affected by the existence of clusters, its finite-size estimates however are biased due to the slow convergence.

left tail of the PDF as "equatorward" extreme energy transports. These extremes are weaker than the median transports, and sometimes slightly negative. The procedure for equatorward extremes is the same as described above for poleward extremes. Hence, we search for a stable shape parameter as function of a decreasing threshold, equivalent to a decreasing fraction of data points below the threshold.

For illustrative purposes, Figure 1 shows the behavior of the shape parameter $\xi$ as a function of the threshold $u$ at different latitudes. We notice that the shape parameter is almost everywhere negative, evidencing that the extreme value distribution is bounded. In other words, Figure 1 provides a graphical explanation of the convergence methodology, justifying the following choice of the thresholds:

- DJF, equatorward: 10% of data below the threshold (10% percentile);

- DJF, poleward: 14% above the threshold (86% percentile);

- JJA, equatorward: 14% below the threshold (14% percentile);

- JJA, poleward: 7% above the threshold (93 % percentile);

We notice that, while $\xi$ is systematically negative for this choice of the thresholds, $\sigma$ is positive and slightly increasing with latitude, denoting an increasingly large spread of extremal values (Fig. 1a, b).

### 2.2.3 Weather regimes

The Weather Regimes (WR) are computed from the daily geopotential height at 500 hPa (z500), using the WRtool Python package (Fabiano et al., 2021). We summarize the methodology here; for more details and discussion the reader is referred to the Methods section in Fabiano et al. (2020). We focus on latitudes between $30°$ and $90°$ N and three different longitudinal sectors: the Euro-Atlantic (EAT, from $80°$ W to $40°$ E), the Pacific-North American (PAC, from $140°$ E to $80°$ W) and the whole Northern Hemisphere mid-latitudes (NH).

The original data were first interpolated to $2.5°\times2.5°$ horizontal resolution; since the WR diagnostic aims at capturing large-scale configurations, this step does not alter the results of the analysis. Before computation, the daily mean seasonal cycle - smoothed with a 20-day running mean - is removed from the data to obtain daily anomalies.

To retain only the large-scale component and reduce dimensionality, an Empirical Orthogonal Function (EOF) decomposition is performed, separately for each considered longitudinal sector. The minimum number of EOFs that explain at least 90% of the total variance is retained: this corresponds to 16, 19 and 41 EOFs during DJF for the EAT, PAC and NH, respectively; for JJA, 22, 25 and 57, respectively. An EOF-based dimensionality reduction is a common first step in weather regimes detection algorithms (Cassou, 2008; Dawson et al., 2012; Cattiaux et al., 2013; Straus et al., 2017; Dorrington et al., 2022). Sensitivity tests on the number of EOFs retained (not shown) confirm that the regime patterns are robust to this choice (e.g. Fabiano et al. (2020)), with the full field recovered in the limit of all EOFs. The fact that higher rank EOFs do not provide a significant advantage in terms of accuracy of the clustering, implies that they denote mainly noise. The WRs are then computed by applying a $K$-means clustering algorithm to the selected Principal Components (PCs). Appendix B highlights that differences in the

distribution of first four PCs between the full population (all days) and the population of extreme events are consistent with the differences observed in the weather regimes frequencies, offering a complementary view. Although EOFs in the two populations can still be significantly different, this suggests that weather regimes are a fundamental representation of the dynamics, and are thus suitable to investigate the differing statistics of the extremes relative to the overall population.

For all sectors and seasons, we set the number of regimes to 4. This choice is widely documented in literature for the EAT and PAC winter circulation (Michelangeli et al., 1995; Cassou, 2008; Weisheimer et al., 2014; Hannachi et al., 2017; Straus et al., 2017), although different numbers of clusters have been used in other studies (e.g. Pasquier et al., 2019; Strommen et al., 2019; Hochman et al., 2021). Each day is assigned to one of the regimes and a set of 4 regime centroids is obtained. The regime pattern is defined as the composite of all anomalies assigned to a certain regime.

Figure 2 shows the pattern of the weather regimes computed for each region (EAT, PAC, NH) and season (DJF, JJA) during 1979-2012, following the procedure described above. DJF patterns (left column) resemble the conventional DJF weather regimes in the literature. These are, for EAT: the North Atlantic Oscillation (NAO) positive phase (hereafter NAO+), the Scandinavian blocking (SC), the Atlantic ridge (AR), the NAO negative phase (NAO-) (Cassou, 2008; Dawson et al., 2012); for PAC: the Pacific trough (PT), the positive and negative phases of the Pacific-North American pattern (PNA+, PNA-), the Bering ridge

(BR) (Jung et al., 2005; Weisheimer et al., 2014; Straus et al., 2017); for NH: the "Cold-Ocean-Warm-Land" pattern (COWL), the PNA-, the Aleutinian ridge (ALR), the Arctic Oscillation negative pattern (AO) (cfr. Kimoto and Ghil (1993); Corti et al. (1999)).

     We refer to JJA regimes (right column) as $xCn$, with "x" being EAT, PAC or NH, and $n = 1, \ldots, 4$ the index denoting the cluster. Thus, for the EAT domain: $EATC1$ features a zonally spread blocking extending from the Eastern Atlantic to Scan-

225 dinavia, $EATC2$ closely resembles the NAO+ pattern, $EATC3$ can be referred to as a Greenland blocking, while $EATC4$ features a dipole structure, with an Eastern Atlantic low and a Scandinavian high (cfr. Guemas et al. (2010); Yiou et al. (2011)). Concerning PAC and NH, all regimes feature anomalies that are much weaker than DJF regimes, given that the circulation as a whole is weaker in the JJA season. $PACC1$ features a weak blocking south-west of Alaska, while $PACC2$ is somehow similar to the BR pattern, although much weaker. $PACC3$ and $PACC4$ appear to mirror each other. $NHC1$ exhibits a Siberian

high, while $NHC2$ features a number of highs surrounding the North Pole; $NHC3$ features a Greenland blocking and $NHC4$ closely resembles $EATC1$.

     As shown in Figure 2, the weather regimes are weaker in summer than in winter, consistent with the seasonal cycle of atmospheric variability (e.g. Faranda et al., 2017). Hence, despite in some cases being apparently similar, weather regimes in the two seasons hint at genuinely different aspects of the dynamics, as further discussed in Sect. 3.2.

For each cluster, the absolute frequency is computed within the population of all events, then the ratio of absolute frequency in the population of extreme events to the absolute frequency in the overall population is obtained, for each tail of the distribution and in each season. Specifically, the relative variations in absolute frequencies are retrieved as $\Delta f'_x = \frac{f'_x}{f_x} - 1$, where $f_x$ is the absolute frequency of occurrence of the cluster $x$ for the overall population, and $f'_x$ is the frequency of occurrence of extreme events in cluster $x$ relative to the total extreme population. Hence a positive $\Delta f'_x$ indicates that cluster $x$ include a

higher fraction of extreme events compared to its fraction of all events, i.e. cluster $x$ is more favourable for extreme events. In

order to assess whether the variation in absolute frequencies is significant, a bootstrapping is performed, through 300 random resamplings of the original population with the same number of events as the number of extreme events in the respective tail. The hypothesis test is performed at the one-sided 95% significance level.

In order to link the weather regimes characterisation of extremal transports to the wavenumber decomposition illustrated in Sect. 3.1, we focus on the zonal wavenumber effecting the largest share of the meridional energy transport anomaly. First, at each latitudinal band, for each assigned weather regime, in each season and for each tail of the distribution, we compute the contribution of each individual wavenumber to the meridional energy transport anomaly with respect to the average across the extreme events in the selected subset. Then, the dominant wavenumber for the specific event is assigned, as the one whose contribution to the transport is largest. Finally, the average of the dominant wavenumbers across the subset of extreme events assigned to the specific regime and occurring at the selected latitudinal band is computed.

## 3 Results

### 3.1 The PDFs of meridional energy transports and their wavenumber decomposition

We start our analysis by looking at the PDFs of the total and wavenumber-decomposed meridional energy transports and their extremes.

The PDFs (filled contour plots) of the total zonally averaged meridional energy transport are computed for each latitude in the 30°–60° latitudinal band, as shown in the top panels of Figures 3 and 4 (for DJF and JJA, respectively). A normalization is performed across latitudes in order to evidence the diversity of ranges at different latitudes. In DJF, the PDF peaks at about 42.5° N, with a value of $6.5 \times 10^{15}$ $W$, the mean of the PDF then decreases towards the higher latitudes. In JJA, the PDF shows a weaker variability across latitudes, with a mean/median value oscillating between 2.0 and $2.5 \times 10^{15}$ $W$, and a moderate decrease towards the high latitudes. PDFs in DJF are overall positively skewed, as shown in Figure 5. The highest skewness (about 0.4) is at the equatorward edge of the midlatitudinal band, while the lowest skewness (0.2) is in proximity of the median peak of the total transport around 45°N. In JJA, the PDF is more skewed north of 40°N, with values ranging between 0.3 and 0.25, whereas it is more symmetric south of 35°N, with values of about 0.1. A positive skewness indicates that the PDFs is asymmetric, with a larger or longer tail of positive ("poleward") than negative ("equatorward") extremes. The skewness values are consistent with the eddy circulation in mid-latitudes favoring intermittent, locally strong poleward transports that also have a signature in the zonally integrated transport (cfr. Messori and Czaja (2015); Marcheggiani et al. (2022)).

The extreme transports for both equatorward and poleward tails of the distributions are identified from the PDFs of total zonally averaged transports, following the methodology described in Sect. 2.2.2. Both sets of extreme transports are heavily skewed in both seasons, with poleward extremes being more skewed than equatorward extremes (Fig. 5). The asymmetry of the total transports is consistent with this slight difference. In the overall PDFs, meridional sections of DJF and JJA skewness are opposed, with larger values at the equatorward edge in DJF, in the centre of the midlatitudinal band in JJA. The larger skewness at the equatorward edge hints at the presence of relatively larger extreme transports, where the non-linear growth of the baroclinic waves is more pronounced (Novak et al., 2015), despite the median transport being weaker than at the centre of

the domain. This is also mirrored in the meridional section of DJF skewness for poleward extremes (bottom, right in Figure 5). Equatorward extremes attain smaller or even negative values at high latitudes, where the mean transport is weaker, which emphasises the importance of such extreme events for setting the seasonal energy transport at high latitudes. The opposite occurs for the poleward extremes, with the largest extremes being stronger for those latitudes where the mean transport is stronger. This is in agreement with Lembo et al. (2019), who found that both poleward and equatorward extremes are related to the mean strength of the transport (cfr. their Figure 1d-g).

We next decompose the transport into its wavenumber components (panels c)-f) in Figures 3 and 4). Similarly, the extreme values sampled from the PDF of the total transports, are decomposed in wavenumbers. In both seasons the most significant contribution to the overall transport comes from the planetary-scale and synoptic-scale contributions. The equatorward and poleward extremes largely overlap in the zonal mean component, and both attain negative values, while the zonal mean component does not provide a significant contribution to the extreme transport. Very different features emerge at the planetary and synoptic scales. During DJF and at both these scales, the magnitude of poleward extremes follows that of the median transport for the respective component, and attains large values when the median component is at its maximum. Equatorward extremes, on the other hand, only partly follow the median curve and attain near-zero values at almost all latitudes, especially regarding the synoptic contribution. During JJA, a significant share of planetary-scale equatorward extremes feature net negative transport in the middle latitudes, where the contribution to the transport is almost vanishing on average. This is consistent with planetary waves being able to act passively, transporting energy counter-gradient, and thus with no conversion of available potential energy into kinetic energy, as discussed in Baggett and Lee (2015); Lembo et al. (2019). Synoptic-scale waves, instead, almost everywhere share the positive sign and the meridional structure of the mean transport, although as in DJF, the most extreme events during JJA also attain near-zero values at almost all latitudes.

To provide a clearer view of the relative importance of the different components, we show in Figure 6 meridional sections of the ratios of zonal (in blue), planetary (in red) and synoptic (in yellow) scales to the total transports and their extremes for the two seasons. The latitudinal gradient is largest in DJF, with the planetary scales becoming relatively more important (up to 1.2 times the total transport) and the synoptic scales decreasing (starting from 0.4 at 30°N to 0.1 at 60N) with increasing latitude. The zonal component, in turn, has the same sign as the total transport in the lower latitudes, then turns negative and opposes the eddy components, having a ratio of up to 0.4 of the total transport. In JJA, the relative ratios of planetary and synoptic component are antisymmetric, and the zonal component is steadily negative, opposing the eddy transport. Extreme transports show similar features, with the ratio of planetary scales in JJA equatorward extremes becoming negative around 45°N, and remaining small at higher latitudes, while the zonal component consistently opposed about half of the total poleward transport at all latitudes. The planetary and synoptic scales thus rarely contribute in a similar way to the overall transport, and this is especially true for the extremes. Indeed, planetary and synoptic extremes are rarely happening concomitantly, as already found in Lembo et al. (2019).

## 3.2 Weather regimes and zonal wavenumbers associated to extreme events

We now shift our attention to the detection of weather regimes in the population of extreme events, in order to investigate the relation between extreme transports and recurrent patterns of the large-scale circulation.

As discussed in Appendix B, conditioning on different extreme transport events yields different average values of the PCs of the leading EOFs, suggesting that the population of extreme events might exhibit a preferred development of specific weather regimes. Following from this hypothesis, we compute $\Delta f'_x$ in the population of extreme events (as described in Sect. 2.2.3), grouping extremes by 2-degrees wide latitudinal bins. Figures 7-8 display $\Delta f'_x$ for each of the chosen regimes in the EAT, PAC and NH regions, for poleward (left) and equatorward (right) extreme events in DJF and JJA, respectively.

DJF poleward extremes are characterized by anomalously negative z500 anomalies in the lower mid-latitudes and high-latitudinal blockings over the Northern Atlantic, as denoted by stronger frequency of NAO- (Fig. 7a) and AO (Fig. 7e) during extreme events. In the Pacific region, PT (Fig. 7c) regime frequency increase denotes large meridional exchanges with wide ridges and troughs. Consistently, a significant reduction in the NAO+, BR and ALR modes is found. In JJA, NHC4/EATC2 (Fig. 8a,e), characterised by a pattern similar to an Eastern Atlantic-Scandinavian blocking, is increasingly likely. EATC4 (Fig. 8a) and PACC4 (Fig. 8c), denoted by negative anomalies in the eastern boundaries of the Atlantic and Pacific oceans, are rarer, as well as NHC3 (Fig. 8e), featuring a weak Pacific ridge and a Greenland blocking. In general, we observe that changes in absolute frequencies are coherent across neighbouring latitudes, consistently with what is observed for the energy transport extremes, often occurring concomitantly at several latitudes (compare to Fig. 1d-g in Lembo et al. (2019)).

DJF equatorward extremes largely mirror what is described for poleward extremes: NAO- (Fig. 7b), PT (Fig. 7d) and AO (Fig. 7f) modes become generally less frequent, whereas NAO+, PNA-, ALR and, to some extent, BR modes are more frequent. In other words, a more zonal North Atlantic regime and stronger Pacific blocking are associated with weaker meridional transports. This is in line with the opposite signs of PC fractional changes described in Appendix B. Looking at JJA, it clearly emerges that the $PACC4$ (Fig. 8d) pattern becomes more frequent, while $PACC3$ (Fig. 8d) and $NHC3$ (Fig. 8f) (especially for extremes at high latitudes) are rarer. Remarkably, $EATC3$ and $EATC1$ (Fig. 8b) become increasingly likely for extremes at low and mid-latitudes, respectively, $EATC2$ for extremes at higher latitudes. This suggests a weakening and a northward shift of the centres of baroclinic activity, as the latitude at which the equatorward extremes are selected increases, although a direct comparison is not provided here.

We then focus our attention to the dominant wavenumber, as a function of the region and of the weather regime to which extreme events are attributed. We remark that the spectrum of the meridional energy transports is now different from the one considered in Figures 3 and 4, as we focus on the wavenumber decomposition anomalies relative to their climatological contributions, rather than on their absolute values. For this reason, we limit our range of wavenumbers to $0-7$, as no higher wavenumber appears to be dominant in any of the considered cases.

No matter in which region the clustering is focused, DJF extreme events are associated with dominant wavenumbers between 2 and 4 (Figure 7). In JJA (Figure 10), the situation is more diversified, with dominant wavenumbers in the range of $k = [2, \ldots, 6]$ and a clearer dependence on the circulation cluster. The dominance of higher zonal wavenumbers is consistent with

the finding that PC fractional changes in JJA occur mainly for higher order EOFs, as described in Appendix B. Poleward extremes generally peak at higher wavenumbers than equatorward extremes. Focusing on $EATC2$ and $NHC4$, we also find that scales bridging the chosen planetary-synoptic threshold ($k = 5$ and $k = 6$) are co-located with latitudes featuring the largest frequency change (cfr. Figure 8). Interestingly, to some extent the same happens for $EATC4$, $PACC4$ and $NHC3$, where the latitudes featuring the largest frequency reductions are also characterised by relatively higher wavenumbers (compared

to extremes in other clusters and in the same cluster but different latitudes), again $k = 5$ and $k = 6$. Dominant wavenumbers for equatorward extremes are less sensitive to frequency variations: events associated with $PACC4$ and $EATC1$ frequency increases are associated with a dominant $k = 4$ wavenumber. $EATC2$ features a $k = 3$ dominant wavenumber in the high latitudes, whereas $EATC3$ is characterised by a $k = 2$ wavenumber in the low latitudes.

## 4   Discussion

The zonal wavenumber decomposition adopted here poses some challenges when trying to relate meridional energy transport extremes to specific atmospheric circulation features. Above, we have adopted weather regimes as a tool to identify such circulation features, and we argue that they span the diversity of patterns associated with these extremes. In order to further examine the population of extreme transport events and identify persistent patterns in some regions of the domain, it is useful to complement the description of weather regimes and dominant wavenumbers with the analysis of composite mean z500

anomalies. We stress that while the composite mean viewpoint is not informative of the intrinsic variability associated with those extreme events, it allows to better frame already evidenced aspects of the circulation associated with extreme transport events. Figures 11 and 12 display composite mean z500 anomalies for meridional energy transport extremes taken at three different latitudes in JJA and DJF, respectively. Table 1 shows that latitudinal variations in the number of events are within 10% of the sample size, ensuring that the anomaly maps in the two seasons have comparable significance. Clearly, DJF and JJA

differ in the fact that the higher zonal variability in the latter is related to a higher dominant zonal wavenumber. The emergence of patterns consistent with the dominant wavenumbers highlighted in Figures 9 and 10, hints at the amplitude of such waves as determining the strength of the transport.

    JJA equatorward extremes are particularly interesting, given that they account for an a priori surprisingly large share of energy transport from the Pole toward the Equator. As shown in Figure 11, lower-latitude extremes are associated with widespread

negative z500 anomalies, both on the main ocean basins and on land. This is consistent with a mainly equatorward energy transport. For higher latitude extremes, a Euro-Scandinavian blocking pattern emerges, consistently with an increasing frequency of the $EATC2$ pattern, with negative z500 anomalies confined at even higher latitudes, and a dominant $k = 2 - 3$ wavenumber pattern. An interpretation of these features is provided in light of what described above (cfr. Figures 8 and 10). At lower latitudes (30–33° N), the increased likelihood of the $PACC4$ regime is consistent with co-located negative anomalies, espe-

cially over the Pacific, and a dominant $k = 4$ wave. Overall, this hints at the role of the planetary-scale contribution, that is particularly relevant when the synoptic contribution is weak, especially at higher latitudes, where the planetary-to-total ratio is positive, as seen in Figures 4 and 6. Looking at the composites, we interpret this equatorward or very weak poleward transport

associated with equatorward extremes as the result of planetary-scale transport acting to confine baroclinic eddies in the high latitudes, and overwhelming the synoptic-scale transport.

JJA poleward extremes also find a relatively straightforward interpretation in the z500 composites. For extremes located in the 30-33 band, widespread negative z500 anomalies are found (although with very different patterns compared to the equatorward extremes), while at higher latitudes several blocking patterns emerge in the channel. Overall, the regime frequency changes (in particular the reduced frequency of the EATC4, PACC4 and NHC3 regimes) are consistent with this pattern, with the mid-latitudinal negative anomalies being replaced by co-located high geopotential ridges. Consistently with Figure 10,

evidencing that these patterns are dominated by wavenumbers $k = 5 - 6$, Kornhuber et al. (2020) find that these modes of the general circulation are associated with the development of co-located heat waves in the NH Summer. We thus argue that heat waves may plausibly be related to the occurrence of extremely strong poleward meridional energy transports.

     To heuristically test this hypothesis, we focus on poleward extreme transports in JJA for the year 2010, which was characterised by a number of concurrent heat waves developing in several regions of the NH mid-latitudinal band, especially

Central-Eastern North America and Russia (Dole et al., 2011). Figure 13 shows analogous results to those shown in Figures 8, 10 and 11, for JJA 2010 only. Extremes at the three chosen latitudinal bands are associated with a positive anomaly over Eastern Europe and Russia, particularly significant in the 45-47 latitudinal band. The composites are also consistent with an increased frequency of the $EATC2$ and $NHC4$ patterns (Figure 13b), especially in the higher latitudinal bands. In addition to that, some frequency reductions, that are not as relevant in the rest of the extreme events population, are found for $EATC1$, $PACC1$ and

$NHC1$, pointing towards a reduced occurrence of a regime similar to a NAO+ pattern and of the so-called "Aleutinian ridge".

     Kornhuber et al. (2020) interpreted the 2010 heatwave as a case of concomitant heatwaves over several regions of the Northern Hemisphere, a consequence of quasi-resonant (QRA) amplification of stationary Rossby waves (Kornhuber et al., 2017; Petoukhov et al., 2013). They found that $k = 5 - 7$ were particularly favorable to the development of this pattern. Consistently, Figure 13c shows that more frequent regimes in the energy transport extremes population are associated with dominant

wavenumbers ranging between $k = 4$ and $k = 6$. It is not our aim to establish here a dynamical linkage between the proposed QRA mechanism and the JJA poleward transport extremes. We rather conjecture that the emergence of concurrent heat waves with a $k = 4 - 6$ pattern may coincide with the emergence of such zonally integrated eddy-driven transport extremes, that deserves further investigation. Overall, we argue that the 2010 event is "typical" of the more general circulation features associated with poleward meridional energy transport extremes in JJA. The fact that the 2010 event captures a rare yet typically

persistent fluctuation of the summer atmospheric fields in Eurasia has been discussed by Galfi and Lucarini (2021) using large deviation theory arguments. Analogously to the 2010 Russian heatwave case, we also found (not shown) that the Mongolia Dzud event occurred in Winter 2010 (Rao et al., 2015; Sternberg, 2018), which was characterized by intense and persistent cold outbreaks throughout central Asia (Galfi and Lucarini, 2021), Overall, the composite mean analysis points to extreme meridional heat transport events being linked to recurrent atmospheric circulation patterns, which would enable the study of

such extreme events using a rigorous formalism borrowed from statistical mechanics.

     Looking at DJF z500 composites (Figure 12), we note that the three latitudinal bands are characterised by a NAO- pattern, when poleward extremes are taken into account. In the Pacific-North American sector, a strong ridge emerges over the American

continent, moving westward and strengthening at 45-47, before weakening at 57-60. The described pattern is consistent with what found in Figure 7, the NAO-, PT and AO weather regimes becoming more frequent for extremes at all latitudes. At the same time, the Pacific dipole structure is opposed by negative anomalies over the Arctic and the high latitudes, consistently with a reduction in high-latitudinal blockings over that region, as denoted by the decreased frequency of ALR and BR regimes. Composites are also consistent with $k = 2 - 3$ being the range of dominant wavenumbers for the DJF poleward extremes, as seen in Figure 9, with a relatively weak dependence on the weather regime assigned to the extreme event under consideration. The pattern for DJF equatorward extremes is to a certain extent reversed with respect to poleward extremes, with a clear NAO+ footprint in the poleward half of the mid-latitudinal band, and a ALR-like pattern for extremes located in the 45–47 ° band. This reflects the fact that the dominant $k = 2 - 4$ wavenumbers do not change for the two classes of extremes, no matter the regime to what the events are assigned (cfr. Figure 9). It is rather the amplitude of ultra-long planetary-scale waves, carrying energy from the Equator towards the Pole for both poleward and equatorward meridional transport extremes (cfr. Figure 3c), that determine the strength and location of the overall baroclinic eddy activity. Overall, we find that the composite maps are a blend of different regional-scale weather regimes, with significant values roughly resembling the pattern emerging from the description of Figure 7. Zonally integrated transport extremes, in this context, reflect local features of the general circulation that are recurring, and are possibly interconnected by the dominance of ultra-long planetary scale waves.

In DJF, planetary and synoptic-scale transport extremes rarely co-occur (as already stressed in Lembo et al. (2019) and above in this work), and the planetary contribution to the extreme transport is less dependent on the median value of the transport than the synoptic contribution. In other words, in DJF the planetary-scale pattern, embedding smaller-scale eddies, is dominant almost everywhere except than at lower latitudes, and the synoptic scale contribution sharply weakens with the median transport as the latitude increases. Synoptic-scale eddies are thus modulated by the strength of the underlying dominant planetary wave, as clearly suggested by the z500 anomalies. On the contrary, the strengths of the planetary- and synoptic-scale contributions are comparable in JJA. The pattern highlighted by the composite analysis is thus a blend of the two competing factors, with synoptic-scales becoming dominant in central latitudes (for poleward extremes) or the two contributions canceling each other almost symmetrically at most latitudes, confining baroclinic activity where the extreme transports occur (for equatorward extremes).

Summarizing, the composite analysis illustrates qualitatively that the extremes in the overall transport are the result of the interference of high and low wavenumbers (see also the discussion in Messori and Czaja (2014) and Messori et al. (2017) on this topic). Such interference occurs in different ways in the two seasons, with the planetary-scale waves generally determining the latitude and strength of the baroclinic activity, but relative role of the synoptic and planetary waves being very different at different latitudes. The minima and maxima of the planetary and synoptic-scale transports are in fact located at the two edges of the mid-latitudinal band in DJF, and at the centre of the band in JJA (cfr. Figures 3c-d and 4c-d). The zonally inte-grated approach does not allow to identify the phases of the most relevant waves, but we provided some qualitative arguments by comparing dominant wavenumbers, weather regimes and composite analysis. For instance, DJF poleward (equatorward) extremes are denoted by increased (decreased) frequency in NAO-/AO/PT regimes and decreased (increased) frequency of NAO+/ALR regimes, that is also partly reflected in the z500 composite means in Figure 12. This is suggestive of a specific

phase of the $k = 2 - 3$ waves, shaping the synoptic-scale baroclinic activity in the population of extremes and determining the increased/decreased blocking frequencies denoted by differences in weather regime frequencies.

## 5    Summary and Conclusions

In this work, we analysed the zonally averaged meridional energy transports in the mid-latitudinal NH band for the DJF and JJA seasons in ERA5 Reanalysis. We decomposed the transports depending on their zonal wavenumbers, and isolated four contributions: zonal mean ($k = 0$), planetary scales ($k = 1-5$), synoptic scales ($k = 6-10$) and higher scales ($k = 11-20$). We applied an EVT-based selection methodology to isolate extreme meridional energy transport events, labelled as "poleward", when the transport is stronger than average, "equatorward" when it is weaker. We adopted a clustering algorithm for the z500 anomalies associated with meridional energy transport extremes, and we looked for dominant patterns and preferred wavenumbers as a function of the latitudes at which the extremes are detected. We used together with an interpretation of geopotential height composites of extreme transport events.

We find that energy transport extremes emerge as a result of the growth of different wave types across the mid-latitudes, particularly synoptic and planetary scales depending on meridional location and season. Specifically, dominant patterns and preferred wavenumbers suggest that planetary scales determine the strength and meridional position of the synoptic-scale baroclinic activity with their amplitude, exhibiting significant seasonal differences. In DJF, they modulate the synoptic-scale activity, being generally stronger than all other smaller scales, while in JJA they chiefly interfere with the latter being of similar magnitude, either constructively (poleward extremes) or destructively (equatorward extremes). Notably, equatorward extremes feature mainly negative-signed planetary-scale transports north of 42° N.

Understanding meridional energy transport extremes is key to identify mechanisms through which diabatic heating and temperature gradients are balanced. We demonstrated that some preferred regimes favour extreme transports, and that they reflect to some extent preferential modes of variability related to specific zonal wavenumbers. This has clear implications for high latitude warming/cooling, as already stressed before regarding the influence of synoptic-scale eddy activity on Arctic weather (cfr. Ruggieri et al. (2020)) or dynamical patterns related to extreme near-surface temperatures over the Arctic (cfr. Messori et al. (2018); Papritz (2020)).

The emergence of dominant zonal wavenumbers associated with extreme meridional energy transports also has implications for teleconnections. Investigations based on large deviation theory have shown that persistent weather extremes are associated with very large scale atmospheric features (Gálfi et al., 2019; Galfi and Lucarini, 2021; Gálfi et al., 2021). We emphasized here the similarity between our results and the concurrent emergence of heat waves across the Northern Hemisphere associated with a persistent $k = 5 - 7$ Rossby wave pattern, as found by Kornhuber et al. (2020). Our results emphasize that the variability modes related to energy transport extremes are hemispheric in scale, and suggest that regional features such as the NAO or PNA trigger co-variability of weather in remote regions (Thompson and Wallace, 1998; Branstator, 2002). This was not clear a priori, as the zonally integrated approach we adopt here does not allow, in principle, to select recurrent localized extreme transports, linked to regionally constrained circulation patterns. Further, analysing the aggregated effect of wave packets on

meridional energy transports and how they manifest in terms of weather regimes provides a thermodynamic background to the concept of QRA (Petoukhov et al., 2013; Coumou et al., 2014; Kornhuber et al., 2017) for the development of weather extremes, potentially linking local events to anomalies in the general circulation. Indeed, Petoukhov et al. (2013) already stressed that a weakened zonal component of high-amplitude waves is associated with the QRA hypothesis. This does not exclude, however, that the QRA mechanism can be associated with extremely strong meridional energy transports. We consider the investigation of dynamical linkages between energy transports, temperature (and moisture) advection in the context of co-recurrent heat waves amplified by QRA mechanism is a potentially relevant topic for a future work.

Here, we estimated the extremal index with the aim to decluster the energy transport time series in order to accelerate the convergence of extreme value statistics. The extremal index itself, however, contains valuable information related to the persistence of extreme states, as it is the inverse of the mean cluster size of extremes (Coles et al., 2001). In future studies, one could concentrate more on the persistence of extreme (and non-extreme) events in meridional energy transport based on the extremal index. The persistence of individual states together with the local attractor dimension can be used as a dynamical proxy of the general circulation. This approach has already successfully been applied to dynamical features of the atmosphere (Faranda et al., 2017; Messori et al., 2021)), but never, to the best of our knowledge, to thermodynamic features.

Finally, a possible outcome of this analysis, that is left for future work, is the study of meridional energy transport extremes, their zonal wavenumber decomposition and the underlying dynamical drivers, in numerical climate simulations. Assessing the statistics of these events against reanalysis-based data would provide an important background for the study of the climate response to external forcing and how changing statistics in extreme events related to baroclinic eddies can affect future weather predictability in the mid-latitudes (cfr. Scher and Messori (2019)).

*Data availability.* ERA5 Reanalysis data are publicly accessible via Copernicus Data Storage. The computation of meridional energy transport and its wavenumber decomposition was performed at the ECMWF data server and stored at the Nird storage facility provided by the Norwegian e-infrastructure for research and education UNINETT Sigma2 under the project NS9063K.

## Appendix A:  EOF patterns for k-means clusters selection

Figures A1 and A2 show patterns of the first 4 EOFs for DJF and JJA, respectively. These have been used to obtain the fractional change in the PC mean described in Figure B1.

## Appendix B:  Shifts in the extreme event PCs

We investigate here changes in the PCs of the population of extremes, compared to the population of all days. For the sake of simplicity, we restrict ourselves to the 4 leading EOFs, computed in DJF and JJA over the three regions of interest: EAT, PAC and NH. Fractional changes in the mean of the PC distribution relative to the climatological standard deviation (Figure B1) are

evidenced, when significant according to Welch's T-test. We investigate shifts in the mean of the distribution in each dimension (PC), relative to the standard deviation of the climatological distribution.

In DJF, significant shifts are found for the first PC, corresponding to the positive phase of the Arctic Oscillation (AO) pattern for NH (Thompson and Wallace, 1998) and to the corresponding local patterns for EAT (NAO+) and PAC (Bering Ridge) (see Fig. A1 in Appendix A). Consistently, the distributions shift in opposite directions for equator- and pole-ward transport
extremes respectively. The distribution of the third PCs (denoting a positive geopotential anomaly over the North Atlantic in EAT and over eastern Siberia in PAC) also change significantly in the population of equatorward extremes. In JJA, the most significant changes occur for higher order EOFs, with the only exception of the poleward extremes in EAT, connected to the first EOF (NAO-). These higher order EOFs are typically associated with localized patterns (e.g. EOF 3 and 4 for EAT and NH in Fig. A2), indicating that the extremes in this season are related to smaller scales of the circulation.

**Appendix C:  Energy transport PDFs for different choices of planetary wave threshold**

The choice of the threshold for the separation between planetary and synoptic scales, as well as the best indicator to consider the different scales, is the topic of an onggoing scientific discussion (Heiskanen et al., 2020; Stoll and Graversen, 2022). Here, we limit ourselves to testing an alternative wavenumber grouping, based on available literature (e.g. Baggett and Lee (2015); Shaw (2014); Rydsaa et al. (2021)). The same wavenumber might be identifying different types of motions at different
latitudes (as discussed in Heiskanen et al. (2020); Stoll and Graversen (2022)), such that a lower threshold may be appropriate att higher latitudes (e.g. k=3 (Rydsaa et al., 2021), k=4 Shaw (2014)). Given that we analyse a relatively broad latitudinal band, our choice of k = 5 is a trade-off to provide a sensible length-scale for the separation at different latitudes. Figure C1 shows the meridional energy transport PDFs for an alternative grouping, namely zonal wavenumbers k=1-3, denoted as "ultra-long planetary waves", k=4-6, as "planetary waves" and k=7-9 as "synoptic waves". The panel k=0, i.e. zonal mean, is left
unchanged. Starting from the DJF season (left panel), it is confirmed that ultra-long planetary waves are dominant, especially in the generation of "poleward" extremes at higher latitudes, whereas other planetary waves are relevant at all latitudes (with roughly homogeneous contribution across latitudes). The contribution of synoptic waves is weaker than for the original choice of the wavenumbuer groups, although comparable to planetary waves, especially in the equatorward half of the mid-latitudinal band. Looking at JJA, the three eddy contributions are comparable, with planetary and synoptic waves mostly contributing
to poleward extremes in the middle of the channel, and ultra-long planetary waves contributing at lower and higher latitudes. Interestingly, ultra-long and planetary waves have a significant part of their PDFs related to equatorward extremes in the negative domain. In other words, we claim that both components transport energy "counter-gradient", in opposition to the total transport. Overall, one might notice that:

– synoptic-scale waves defined in this way are remarkably homogeneous across latitudes and seasons, so that the only
appreciable change is in the position of the peak. This is broadly coherent with our approach, considering k=6-10 as the synoptic wave domain;

- ultra-long planetary waves play a dominant role in shaping the extremes in DJF, consistent with Figures 7 and 9. Similarly, JJA extremes are characterized by the coexistence of comparable planetary and synoptic contributions, although the former still dominate poleward transports, while ultra-long waves hardly distinguish between poleward and equatorward extremes;

- the regrouped wavenumbers support the conclusion that the strength of the extremes is in all cases dependent on the shape of the median meridional section. The k=1-5 grouping showing no correlation with the median is the result of the latitudinally homogeneous median in the k=4-6 range, plus the weaker contribution by ultra-long waves in low latitudes;

*Author contributions.* VLe and GM designed the analysis and wrote the manuscript. RG performed the wavenumber decomposition of meridional energy transports, VMG conceived the algorithm for extreme events detection and carried out the convergence analysis, FF performed the EOF and k-means clustering analysis and attribution of events. All authors contributed to the interpretation of the results.

*Competing interests.* The authors declare that they have no competing interests.

*Acknowledgements.* G. Messori has received funding from the European Research Council (ERC) under the European Union's Horizon 2020 research and innovation programme (Grant agreement No. 948309, CENÆ project). V. Lucarini acknowledges the support received from the EPSRC project EP/T018178/1 and from the EU Horizon 2020 project TiPES (grant no. 820970). The work is also associated with the Norwegian Science Foundation (NFR) project No. 280727.

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

**Table 1.** Sample sizes for extremes in each of the latitudinal circles chosen for the composite mean maps in Figures 11 and 12.

|        | DJF      |             | JJA      |             |
|--------|----------|-------------|----------|-------------|
|        | poleward | equatorward | poleward | equatorward |
| 30-33  | 1883     | 1483        | 1214     | 2237        |
| 45-47  | 2052     | 1589        | 1148     | 2260        |
| 57-60  | 2114     | 1501        | 1284     | 2452        |

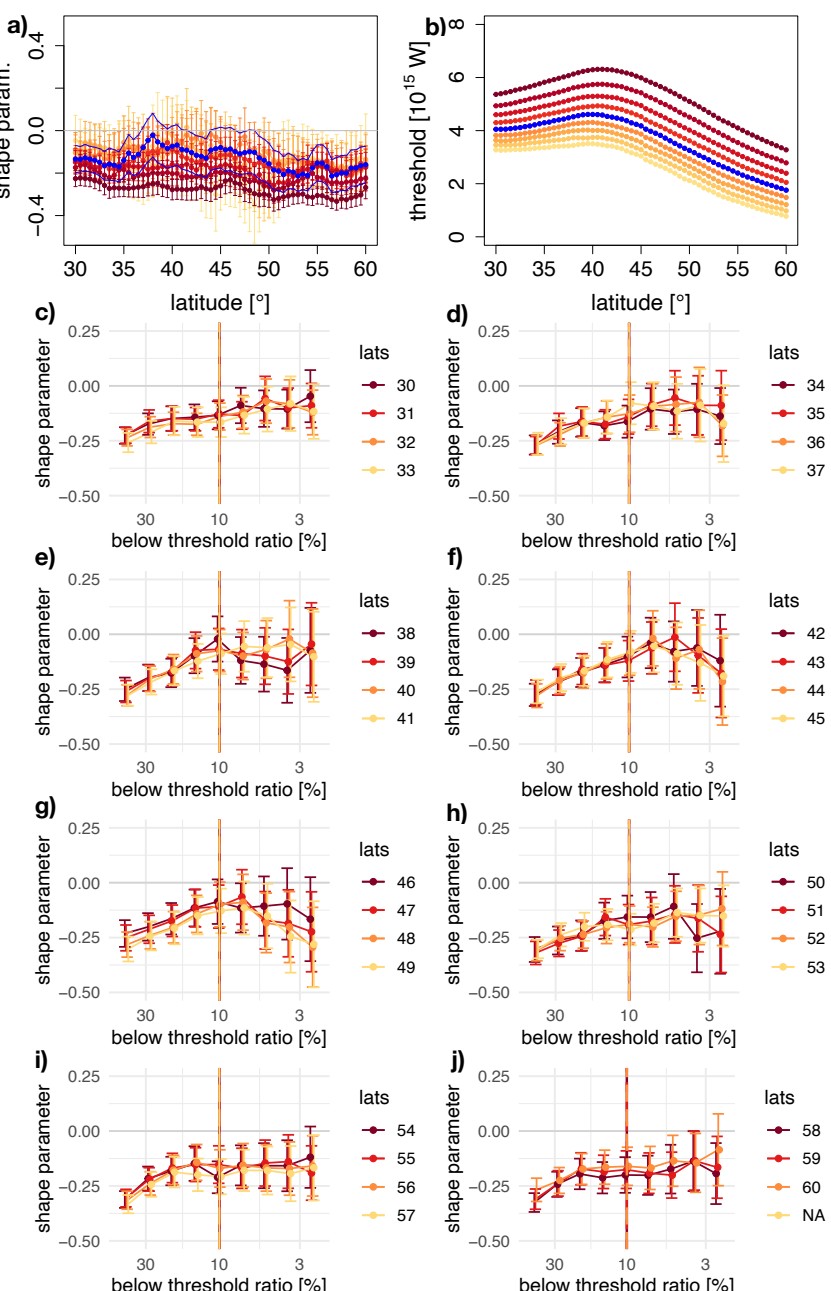

**Figure 1.** Meridional section for equatorward DJF meridional energy transport extremes of: a. Shape parameter $\xi$ (non-dimensional); b. Threshold value [$10^{15}$ W]. The colours in a. denote parameter estimates corresponding to the threshold values shown in c. The blue lines denote the final shape and scale parameter estimates based on the blue threshold values. c-j. Shape parameter as function of threshold (in terms of fraction of data points below the threshold) at different latitudes. The uncertainty range is defined based on 95% maximum likelihood confidence intervals. The orange vertical line denotes the selected fraction of data for the specific tail, which is chosen to be the same at all latitudes.

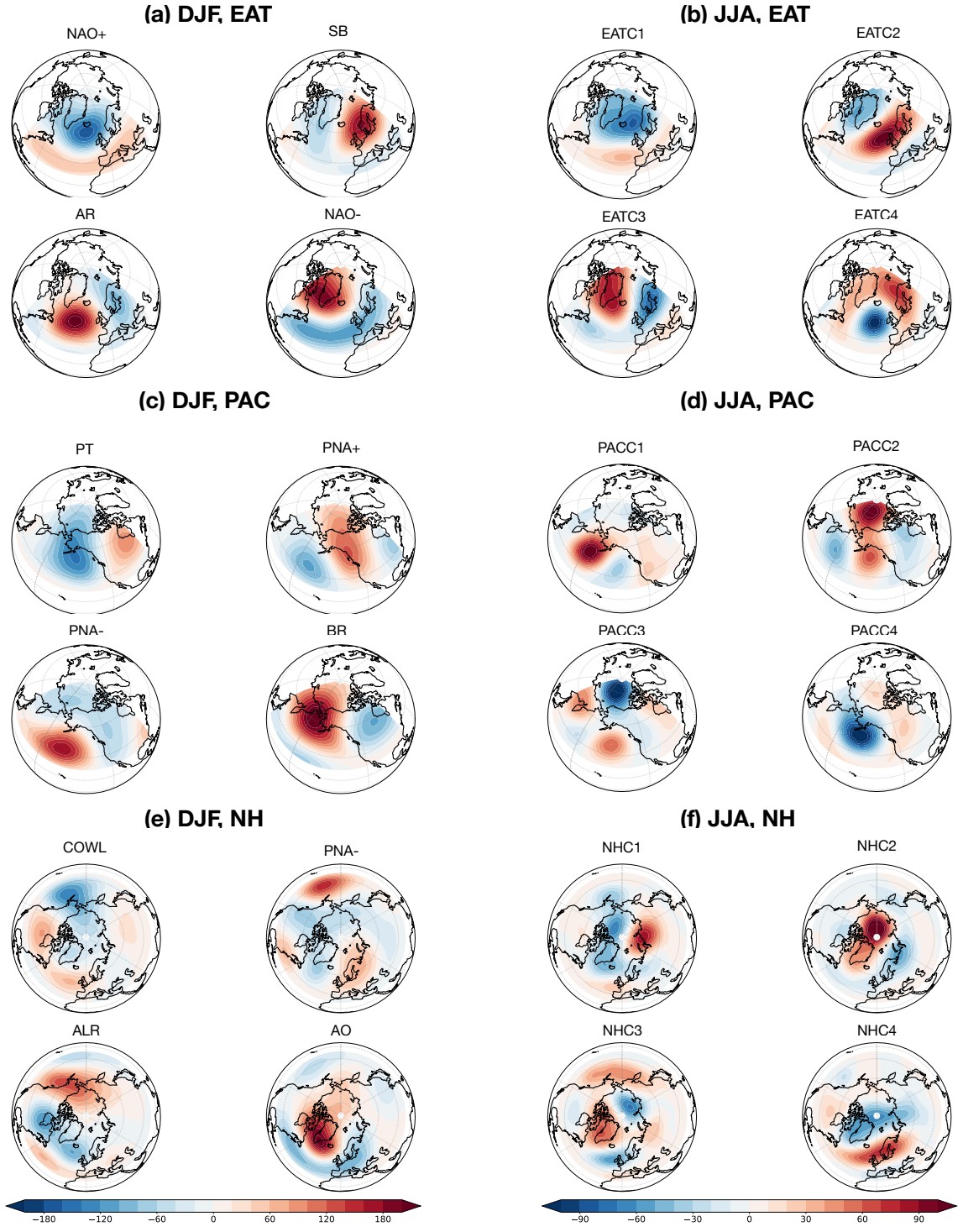

**Figure 2.** Clusters of zg500 geopotential height anomalies (in $dam$) by frequency in: (a) DJF, computed over the EAT region; (c) DJF, computed over the PAC region; (e) DJF, computed over the NH region; (b) JJA, computed over the EAT region; (d) JJA, computed over the PAC region; (f) JJA, computed over the NH region.

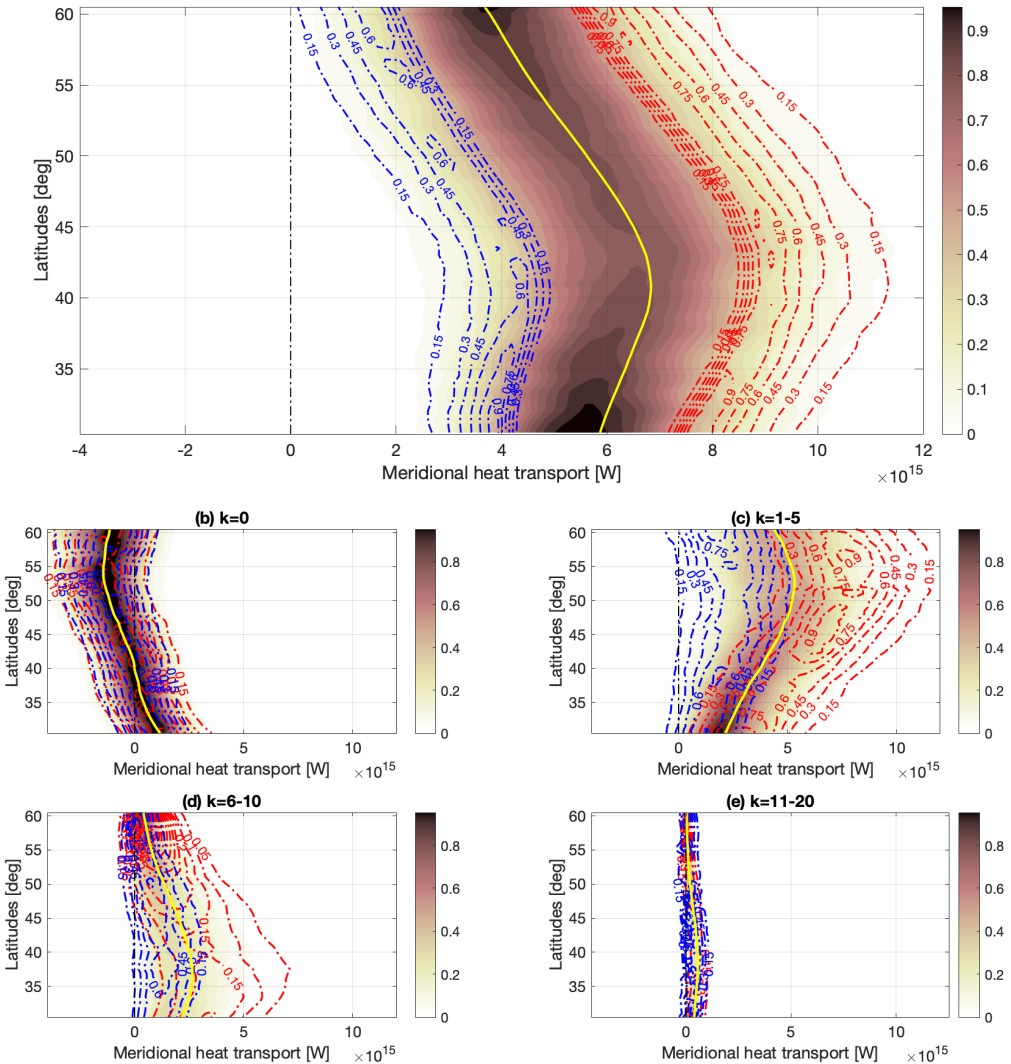

**Figure 3.** PDFs of total (filled contours) and extremes poleward (red contours) and equatorward (blue contours) DJF meridional energy transports over the 1979-2012 period in ERA5. (a) Sum of all wavenumber contributions; (b) k=0 (zonal mean); (c) k=1-5 (planetary scales); (d) k=6-10 (synoptic scales); (e) k=11-20 (higher scales). PDFs are normalized dividing by the maximum value across all latitudes. In order to account for the different range of extremes at different latitudes, the Friedman-Diaconis rule (Freedman and Diaconis, 1981) is first applied to determine the correct number of bin elements for the discretized PDF, then the kernel smoothing estimate (Bowman and Azzalini, 1997) is computed. For graphical purposes, at each latitude the obtained PDFs for the extremes are interpolated on the same number of bins as the PDFs of the overall population.

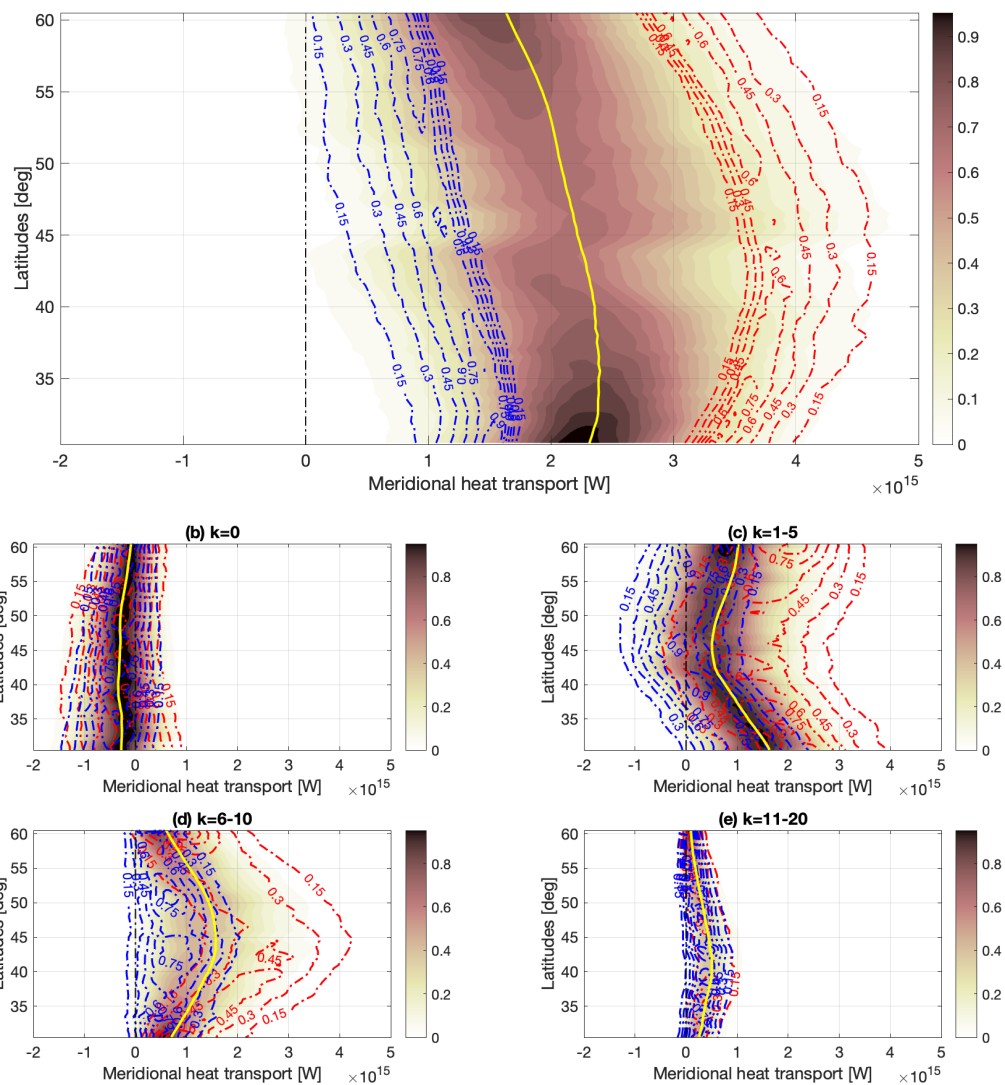

**Figure 4.** Same as in Figure 3, for JJA.

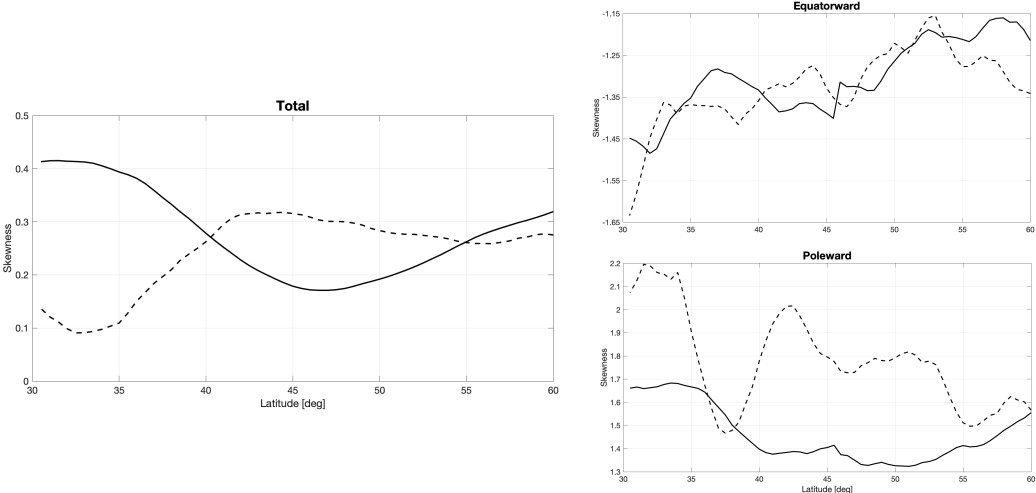

**Figure 5.** Unbiased estimates of meridional energy transport skewness, as a function of latitude, for (left) PDF across all events, (top, right) equatorward extremes only, (bottom, right) poleward extremes only. Solid lines denote DJF, dashed lines JJA.

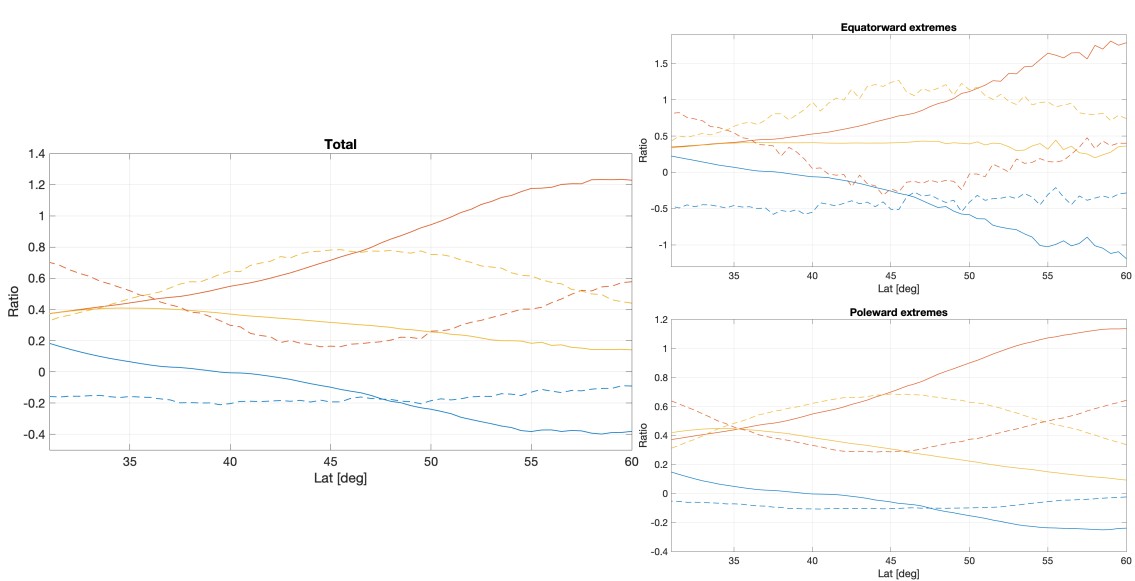

**Figure 6.** Ratios of zonal (blue), planetary (red) and synoptic (yellow) wavenumber components for all events (left), equatorward extremes (upper right), and poleward extremes (lower right). Solid lines denote DJF, dashed lines denote JJA.

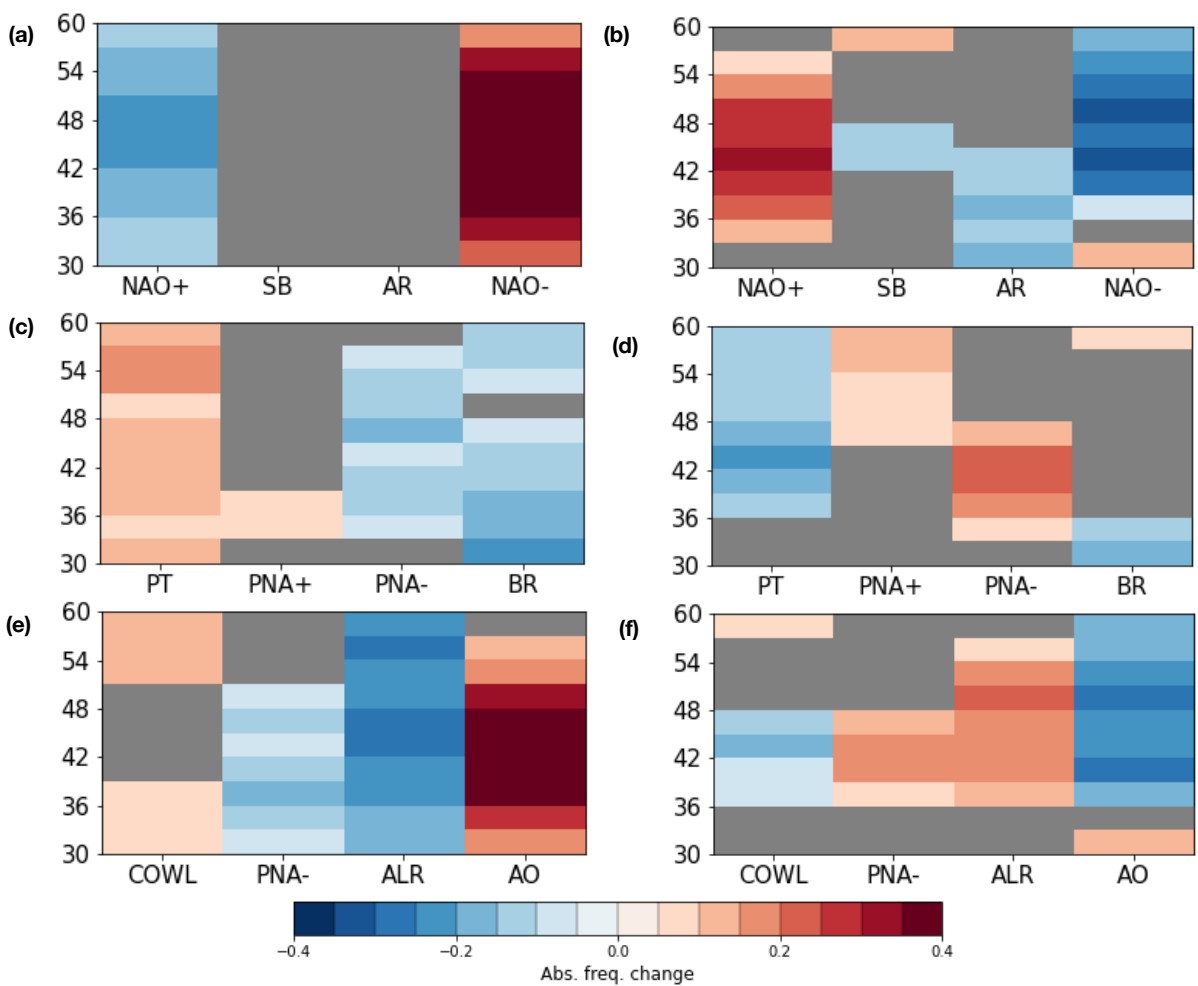

**Figure 7.** Relative variations in absolute frequency of clusters $\Delta f_x'$ in the population of DJF extremes as a function of the latitude at which extremes are found: (a) EAT region, poleward extremes; (b) EAT region, equatorward extremes; (c) PAC region, poleward extremes; (d) PAC region, equartorward extremes; (e) NH region, poleward extremes; (f) NH region, equatorward extremes. Grey boxes denote non-significant frequency variations, according to the bootstrapping methodology described in the Methods section.

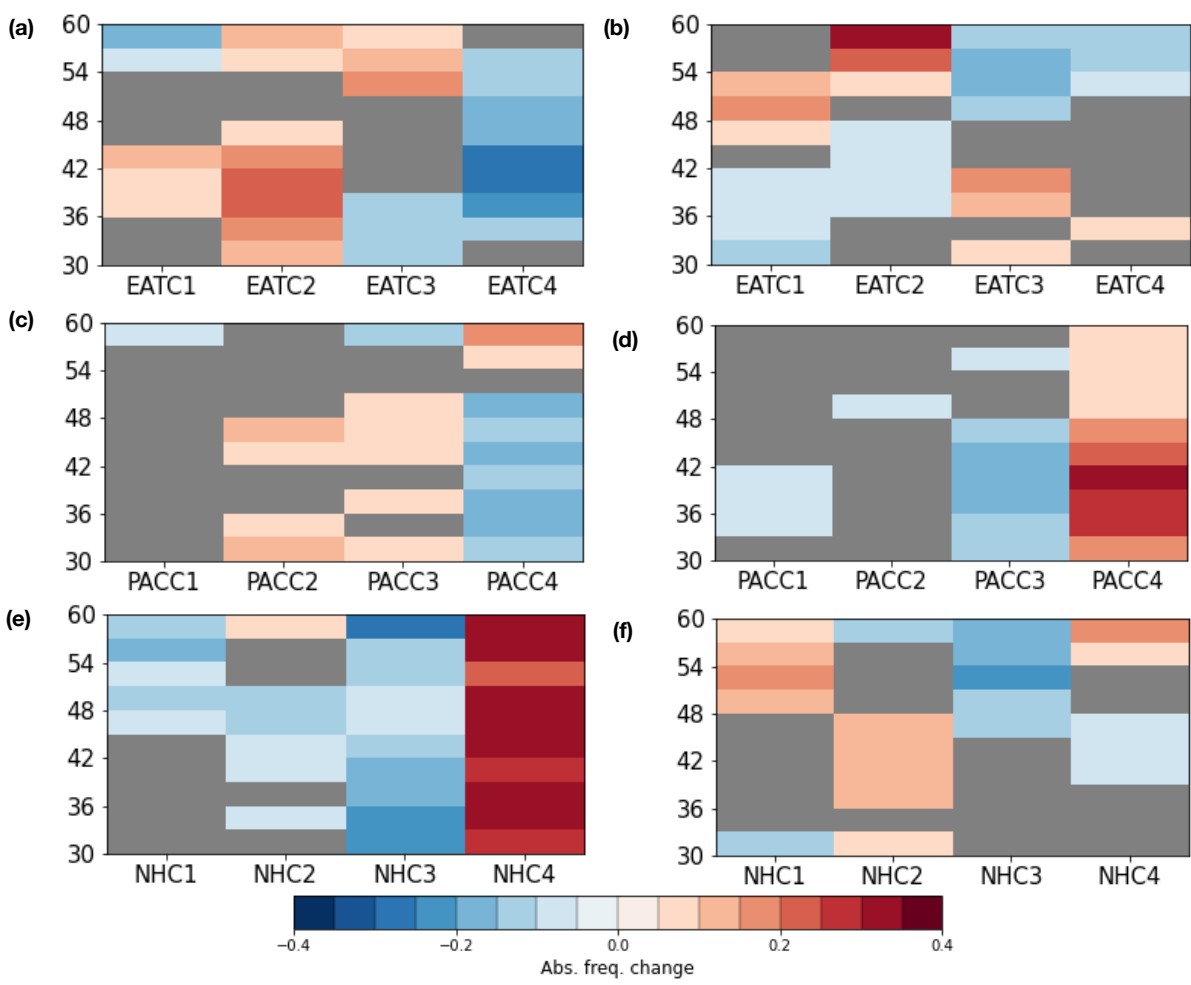

**Figure 8.** Same as in Figure 7, for JJA.

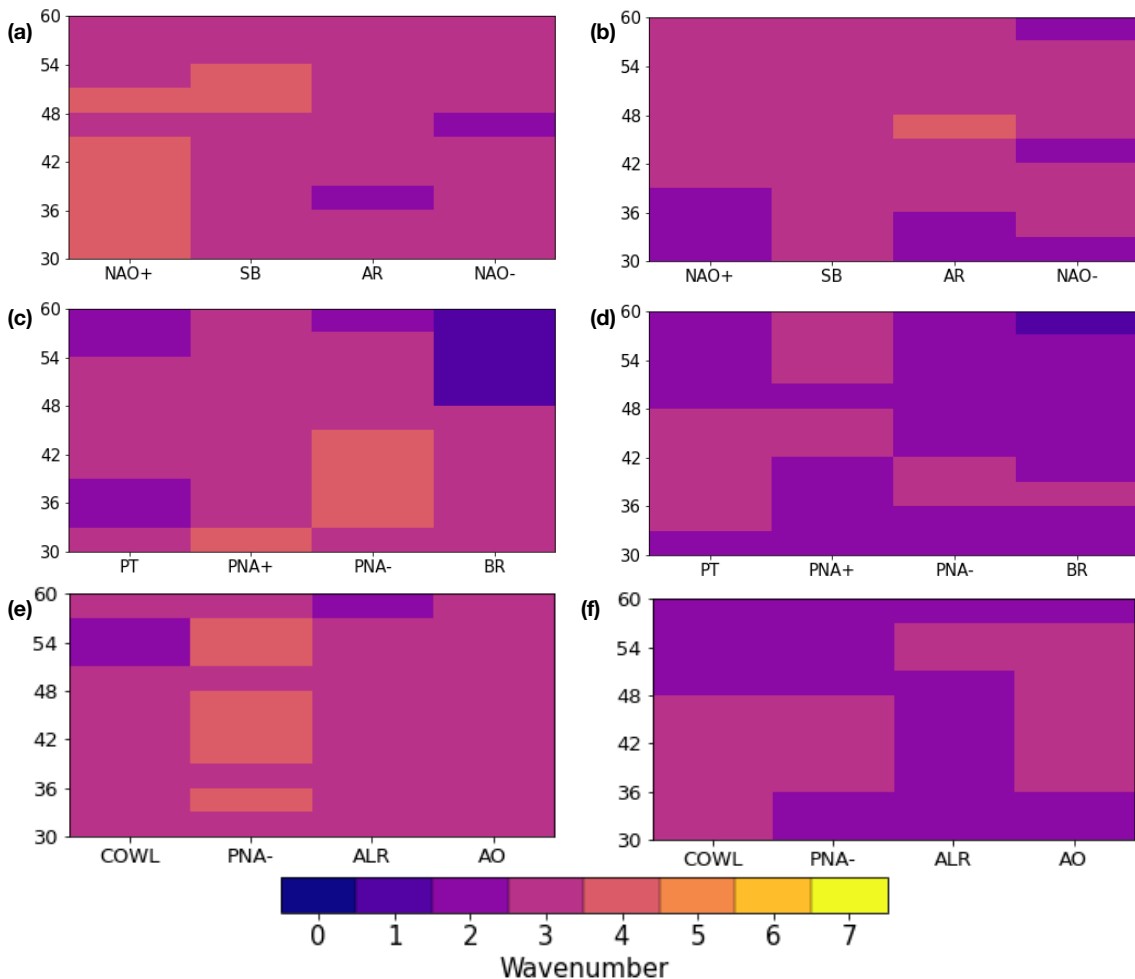

**Figure 9.** Time-averaged zonal wavenumber associated with meridional energy transport extremes (see text) for DJF clusters obtained in different regions, as a function of the latitude at which the extreme is found. (a) EAT region, poleward extremes; (b) EAT region, equatorward extremes; (c) PAC region, poleward extremes; (d) PAC region, equartorward extremes; (e) NH region, poleward extremes; (f) NH region, equatorward extremes.

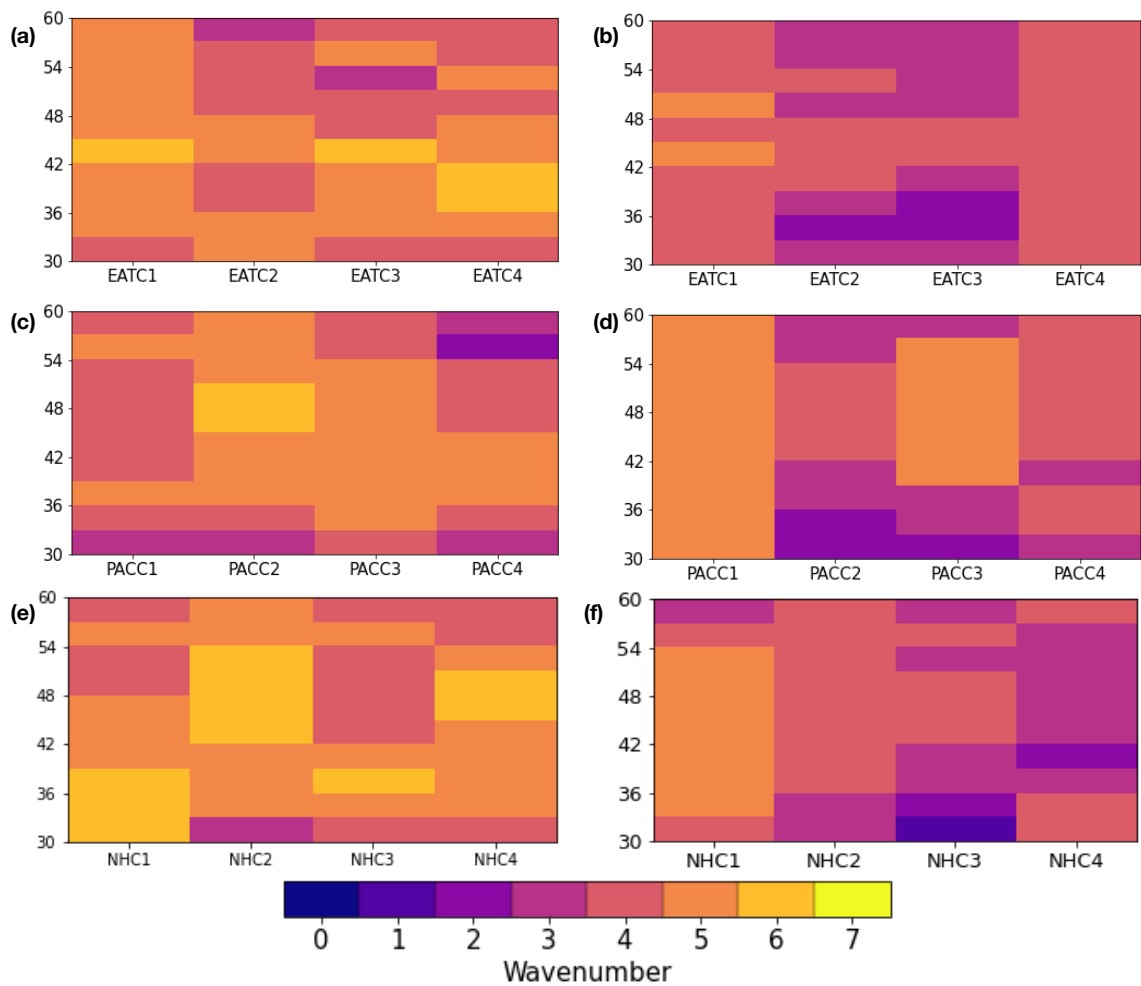

**Figure 10.** Same as in Figure 9, for JJA.

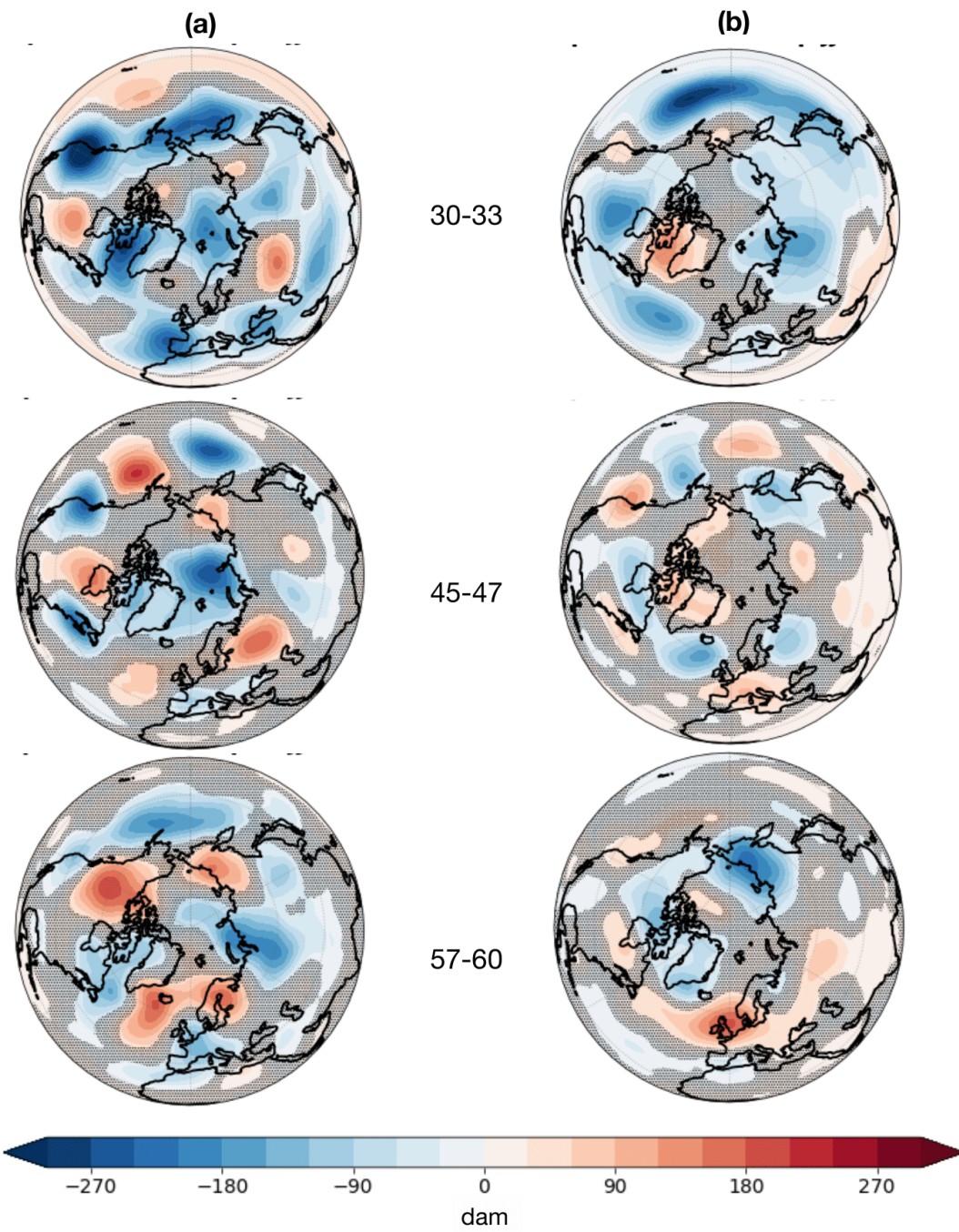

**Figure 11.** Composite mean of z500 anomalies (in $dam$) for JJA (a) poleward and (b) equatorward meridional energy transport extremes at three chosen latitudinal bands. Non-significant values are shaded in grey. The significance is tested wih a bootstrapping method at the 95 % significance level. The null hypothesis is that the composite mean value lies within the range of internal variability.

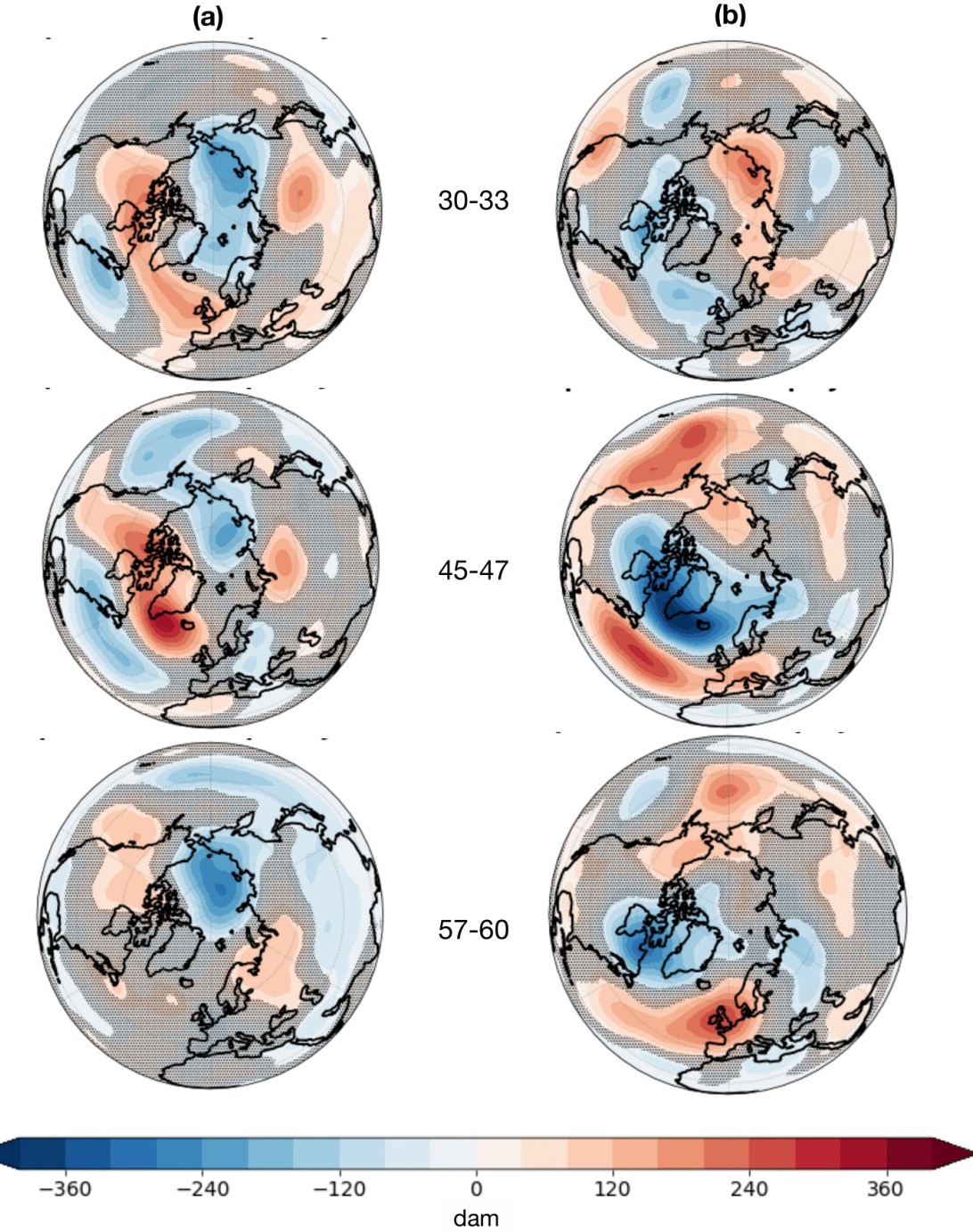

**Figure 12.** Same as in Figure 11, for DJF.

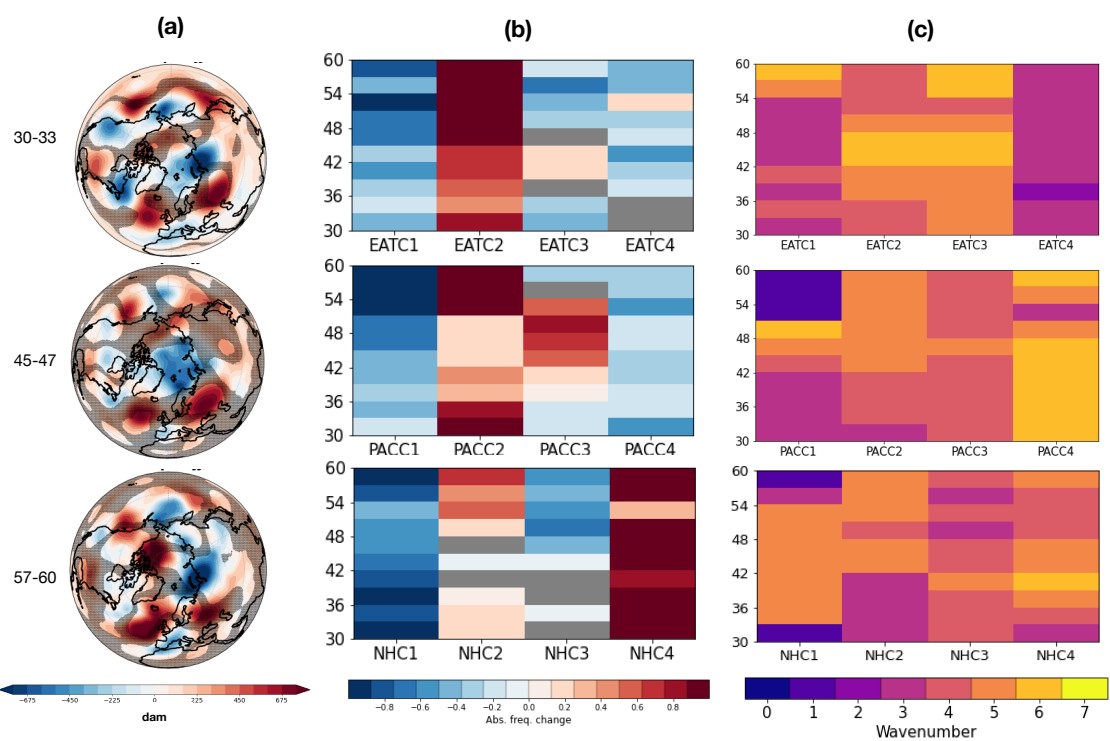

**Figure 13.** (a) Same as in Figure 11, for 2010 poleward extremes only. The extremes are computed with respect to a detrended JJA mean in the 1979-2012 period; (b) same as in Figure 8, for 2010 poleward extremes only; (c) same as in Figure 10 for 2010 poleward extremes only.

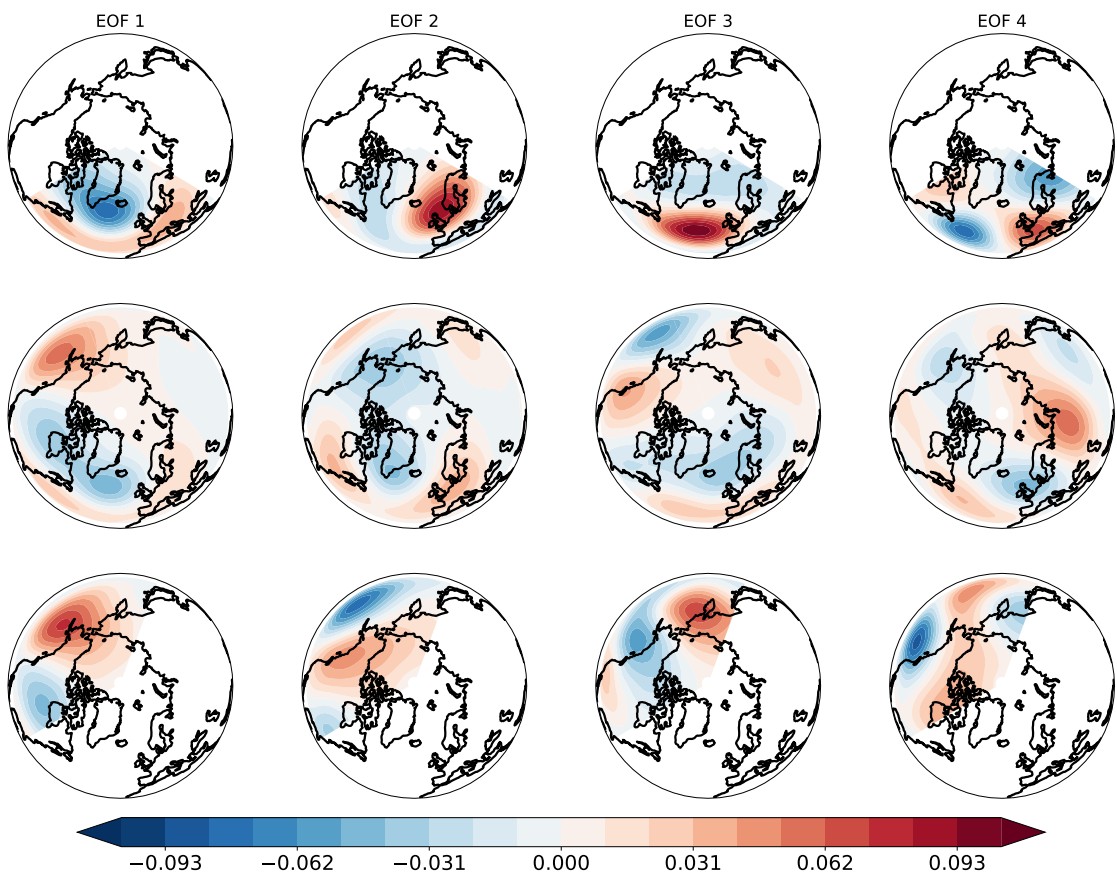

**Figure A1.** Patterns of the 4 leading EOFs for DJF: EAT (first row), NH (second), PAC (third).

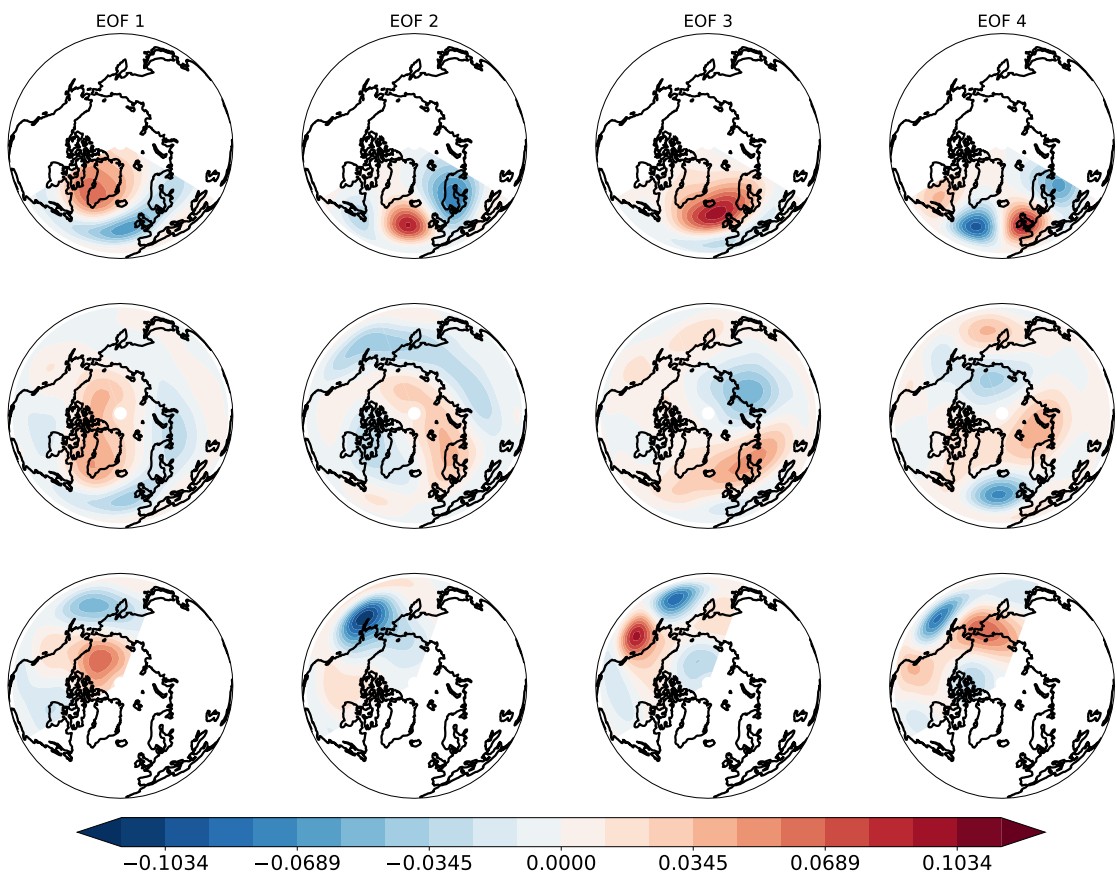

**Figure A2.** Patterns of the 4 leading EOFs for JJA: EAT (first row), NH (second), PAC (third).

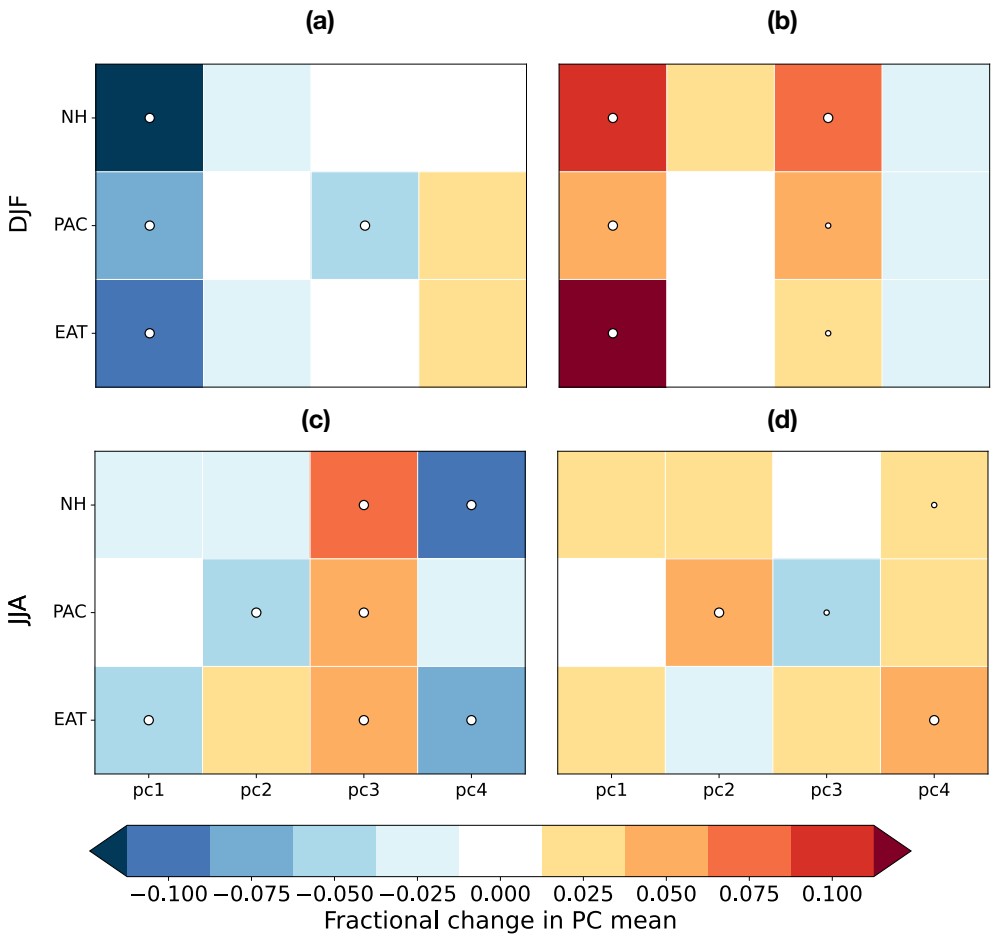

**Figure B1.** Fractional change (relative to climatological std. dev.) of the PC mean for the first 4 leading EOFs, as functions of the region of interest, for (a) DJF poleward extremes, (b) DJF equatorward extremes, (c) JJA poleward extremes, (d) JJA equatorward extremes. White dots denote the significance of the change wrt the overall population of events. Large dots denote 99% significance, small dots denote 95% significance.

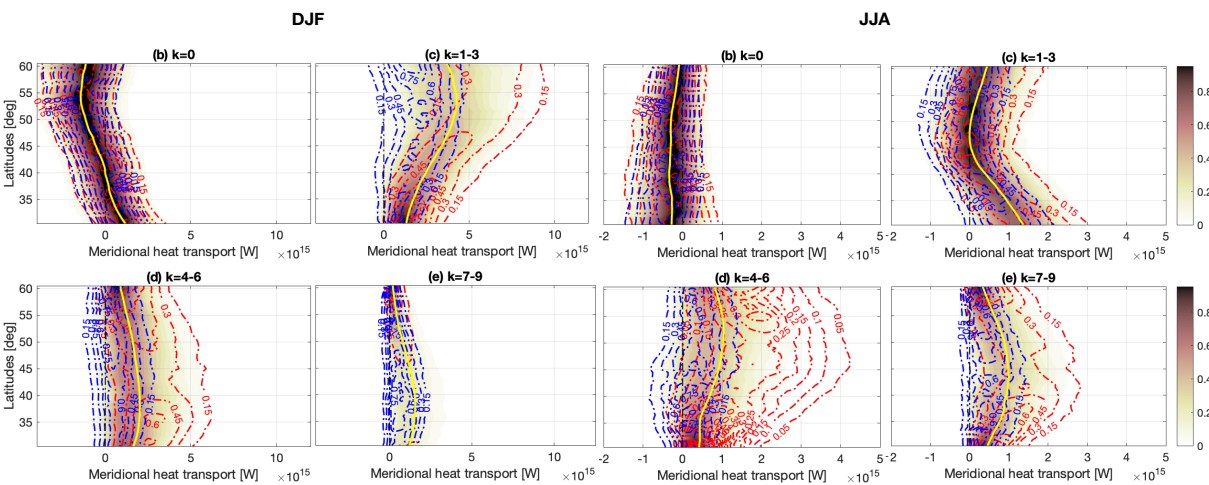

**Figure C1.** Same as in Figure 3b-e, for a different wavenumber grouping, in DJF (left) and JJA (right).