# Peer review of "Meridional energy transport extremes and the general circulation of Northern Hemisphere mid-latitudes: dominant weather regimes and preferred zonal wavenumbers"

_Weather and Climate Dynamics, 2021_

## Author Comment (AC1)

**Reviewer 1**

This paper presents a systematic analysis of extremes in the zonally averaged meridional heat transport and how they are related to weather regimes and preferred zonal wavenumbers. The work is based on several decades of reanalysis data and draws on the results from earlier publications. Overall, the authors argue that their results are consistent with previous results regarding weather regimes, dominant wavenumbers, and how they are related to heat transport extremes. The current analysis makes explicit the role of planetary versus synoptic scales in this context. The question of extreme events is of primary importance in our science, and a detailed analysis such as the present one is welcome. I can see this as a publication in WCD.

Yet, I have a few issues. To be sure, I need to say that I am not an expert in the present topic, rather I consider myself as representative of a typical reader of WCD. As such, I had a hard time in several of the more technical sections to understand what the authors have really done. This is probably due to the fact that the text seems to be primarily directed at the expert, who is familiar with a string of earlier publications from the same group. I have no doubt that the analysis is performed in a proper way; however, I suspect that this is hard to appreciate by the average reader of WCD.

As a way out I suggest that the authors should make a serious attempt to more pedagogically introduce the concepts used in their analysis as well as in their results sections. Instead of just providing the references to multiple generations of previous publications, assuming that each reader is familiar with those papers, the authors should add some advice to the not-so-expert reader trying to introduce and/or summarize these earlier developments on a conceptual level. This would increase the readability of and add great value to the paper.

In addition, the paper would benefit if the authors could add some non-technical guidance to the reader as to what these results mean in more meteorological terms and what the implications are. To be sure, you draw a few interesting conclusions. However, you should make a more serious attempt to connect these conclusions to the more technical parts of the paper. Again, I do not doubt the validity of the results or the conclusions; I just feel that this paper would make a much stronger impact if such meteorological guidance were available and if the technical and the interpretatory parts of the paper are connected in a more seamless fashion. Also, you often point out the consistency with earlier results, and by doing so some readers may get lost and left unclear about what is really new about this paper; therefore, it would be good you could point out more explicitly what is new in the current paper.

We wish to thank the reviewer for their careful assessment of our work, insightful comments and constructive criticisms. We acknowledge that the technical description of the methodology is somewhat difficult to follow for a non-expert reader. In the revised manuscript, we will address the details of the methods in an appendix, in order to ensure reproducibility, and focus more explicitly on the qualitative interpretation of the methodology in the main text. We will further restructure the discussion to distinguish between the results that confirm previous analyses and the results which lead to novel conclusions. .

Examples

Let me provide a few examples illustrating the major issue made above. As I said, some work for improvement would be appreciated in the interest of a broader readership.

For instance, equations (3) and (4) were unclear to me at my first reading. If you do a Fourier decomposition of a field and multiply two such fields (as you have to do to compute a heat flux), you obtain a double sum, one for each expansion. You can, then, sort this double sum according to the resulting zonal wavenumber, and this results in each Fourier coefficient of the heat flux being a sum of many terms from the individual terms (v and E) that just happen to add up to the zonal wavenumber in consideration. This is what I would have expected in equations (3) and (4), but your method is different.

To be sure, I could have read the quoted papers in order to educate myself (to be honest, a cursory look into Graversen and Burtu 2016 did not help me a lot), but I would not be too optimistic regarding the readiness of the average WCD reader to do so. Instead, I would have appreciated not just a short "summary" of those earlier methodological developments, but rather a conceptual introduction on a somewhat higher "meta-level".

In the end, the point here is that you consider zonally integrated fluxes, and Parseval's theorem allows one to express the zonal integral of a quadratic quantity as a single sum over all wavenumbers like in (4). The other important point here is that the sum of all individual components such as (3) and (4) is equal to the total, zonally integrated heat flux, which you refer to as "wavenumber decomposition" later in your text. Implicitly, you heavily draw on this property in the rest of the paper. A corresponding hint in the method's section would have helped me a lot!

The reviewer correctly points out that the Fourier decomposition method has limitations, that our analysis is constrained by consideration of zonally integrated fluxes, and that this has not been sufficiently brought up, neither in the introduction, nor in the Methods section. This  caveat explains why the transports have to be interpreted in hemispheric budgetary terms. Further, as the extreme detection, being zonally integrated, does not give information on localized features of the dynamics, we start our analysis from weather regimes identified in several regions. This is complemented by looking at the composite means of geopotential anomalies. Therefore, we will expand our introduction, better framing the context of our wavenumber decomposition, and split as discussed above the methods section, with a more interpretative/qualitative description evidencing assumptions and caveats of the analysis in the main and an extended technical description in an Appendix , .

To provide a second example, in Fig. 4b it was not clear to me at first why the extremes do not just represent the tails of the distribution from the color fill (just like in a box-and-whisker plot). This is what I would have expected initially. The same problem arises in the text on line 232: how can possibly the "equatorward and poleward extremes largely overlap"? Shouldn't the extremes represent opposing ends of a PDF? If so, it is hard to see how they can overlap. The solution to this problem probably depends on how you defined the extremes and their PDF: the extremes are defined without reference to a wavenumber, and this implies that the existence of an extreme does not have to be reflected in the PDF of each and every wavenumber. Is that right? Other readers may have a similar problem, and some explanation would be very helpful. In addition, reading this (and related) plots is made more difficult due to the fact that the caption does not give contour intervals for the dashed isolines.

We agree with the Reviewer that it is not clear from the current text how the meridional energy transport extremes have been selected. This is done on the population of total heat transports. As a consequence, the PDFs of the wavenumber contributions to the extremes in the two tails of the

distribution can overlap. We will state it more clearly at the beginning of section 3.1, and we will change Figures 3 and 4 in order to account for the contour intervals of the extreme events PDFs.

In the last section, you draw some interesting conclusions, which I was not always able to relate to the core of your analysis. For instance, you say that "planetary scales determine the strength and meridional position of the synoptic-scale baroclinic activity with their phase and amplitude": where exactly have you shown this? How can you make statements about the wave's phase, which (as far as understand) is unavailable from just looking at the zonally integrated heat transport? Similar reservation I have with the conclusions on lines 371-373. I feel that you need to tell the reader somewhat more explicitly how you arrive at these conclusions and which part of the analysis your conclusion is based on.

The zonally integrated approach does not allow to explicitly address the phase of the waves, as correctly pointed out by the reviewer. However, the combined view of dominant wavenumbers, weather regimes and, to some extent, composite analysis, allows us to infer qualitatively how scales interact when total meridional heat transport extremes occur. We argued, in particular, that DJF extremes are the result of a planetary-scale modulation of synoptic-scale eddies, given that:

- the planetary-scale component is the dominant feature of DJF transports, especially north of 40N, with the two tails of the extremes well separated (Figure 3) and previous evidence (Lembo et al. 2019) that synoptic-scale and planetary-scale component extremes are rarely co-located;
- k=2-3 are the dominant zonal wavenumbers for the transport anomalies, as shown in Figure 7, no matter what the sign of the extreme is;
- as Figure 5 shows, poleward (equatorward) extremes are denoted by increased (decreased) frequency in NAO-/AO/PT regimes and decreased (increased) frequency of NAO+/ALR regimes. This is also partly resembled by Figure 10, where z500 composite means are shown;

Based on this we qualitatively argue that planetary scales modulate synoptic-scale baroclinic activity in the population of extremes, with the largest changes in weather regimes related to increased/decreased blocking frequencies and such ultra-long planetary-scale waves dominating the heat transport in the population of extreme heat transports. We plan to rephrase ll. 340-341 and ll. 346-347, l. 371 accordingly, avoiding referring to the phase of the waves.

Take another example: you say on line 385 that "...our results emphasize that the modes related to energy transport extremes are hemispheric in scale". What part of your analysis is this statement based on? My point here is that the chief instrument in your analysis is the investigation of the zonally integrated heat flux, and this leads (almost by design) to "modes" that can be expected to be hemispheric in scale rather than very local or small-scale. In summary, all of these conclusions may be well justified, it was just not easily visible for me. The authors should make an attempt for improvements in this direction.

As above, we acknowledge that the zonally integrated approach somehow limits the opportunity to observe regional-scale features of the meridional heat transport extremes across the NH mid-latitudes. The rationale behind looking at weather regimes in relation to these occurrences was that there could be in principle some mid-latitude regions where the atmospheric circulation is more sensitive to excited meridional heat transport. Looking at the changing occurrences of preferred weather regimes, together with dominant zonal wavenumbers, we found that this is probably not the

case, and that the peculiar role of planetary scales may influence several regions at the same time, as also suggested by the consideration of the 2010 Russian heat wave case. Once again, the significance of composite means (see reply to reviewer 2) complements this. In the revised text, we will clarify the logical reasoning that brought us to draw the conclusion stated on l. 385, and the fact that it is based on a qualitative evaluation of our results. This conclusion is consistent with the referenced works by Comou, Petoukhov, Kornhuber on quasi-resonant amplification (QRA), and we hope that we can expand on the dynamical linkages between co-recurrent blockings and meridional heat transports in a future work.

Minor comments

Line 16: This is somewhat advanced material for the start of an introduction. Presumably you talk about vertically averaged moist static energy, right? In the tropics the vertical change of moist static energy is close to zero, because the increase of potential temperature with altitude is, to a large extent, compensated by a decrease of water vapor mixing ratio.

Agreed. We believe it is sufficient, in this context, to refer to it as simply "heat".

Line 125: you remove the linear trend only in certain latitude bands. Why does this not create awkward discontinuities at the boundaries of these ranges?

The conditional removal of the linear trend was accomplished with the sole purpose of a rigorous application of the EVT-based selection algorithm, particularly to the convergence analysis of the percentile threshold. Thus, it was not applied to the subsequent analysis.

As this was also brought up by reviewer 1, we show in Figure R2 how the thresholds look like if the trend and seasonal cycle are removed everywhere. Whereas the detrending alone (not shown) does not noticeably change the results, the deseasonalizing has a slight effect in DJF, as it can be seen in the first two rows of the figure. There is (left) a small discontinuity at latitude 45°, coinciding with the latitude north of which no deseasonalization has been originally performed. This small discontinuity disappears when the transports are deseasonalized at each latitude (right). However, the change is really minor, especially for the selected thresholds marked by the blue dots, thus it does not affect our results. In case of JJA, there is no noticeable change between the two procedures (left vs. right).

[Figure]

**Figure R1:** Meridional section of threshold values for meridional energy transport extremes selection considering different percentiles (the selected threshold is highlighted in blue, as in Figure 1 of the manuscript). In the left column, transports have been deasonalised and detrended only where necessary, in the right column everywhere: (1st row) DJF, poleward, (2nd row) DJF, equatorward, (3rd row) JJA, poleward, (4th row) JJA, equatorward.

I suggest to increase the size of the panels in Fig. 1 and 2.

Agreed, we will do this.

Panel 1c, y-axis-label: the threshold should have dimensions, right?! How about the physical dimensions of the scale and the shape parameter?

The shape parameter is a non-dimensional parameter and the scale parameter has the dimension 10^15 W; we will state it more clearly in the figure caption.

Fig 3 and 4: How did you normalize the PDFs? It seems to me that integrating by eye over the heat transport at a fixed latitude one may obtain values larger than 1. Or put the other way: what units does the plotted PDF have? Is it really (10^15 W)^(-1)? How should I read the red and blue dashed contours corresponding to the extreme situations (no contour interval given….).

PDFs are normalized at each latitude by their maximum value, so that their maximum value is 1. In order to account for the different number of extremes at different latitudes, the Friedman-Diaconis rule (Friedman and Diaconis, 1981) is first applied to determine the correct number of bin elements for the discretized PDF, then the kernel smoothing estimate of the PDF (Bowman and Azzalini 1997) is computed. A preliminary condition is applied on the number of bins, that has to be at least 2. PDFs are then normalized at each latitude by their maximum value, so that their maximum value is 1. For graphical purposes, the obtained PDFs for the extremes are finally interpolated on the same number of bins as the filled contour plots of the overall population, which is the same at all latitudes. All this will be detailed in the caption to Figure 3 of the revised manuscript.

Line 217: "…. the PDF steeply decays towards the high latitudes….", I understand what you want to say, yet, it is not really well expressed. You probably want to say that the mean or median of the PDF decreases as one goes to higher latitudes.

That is indeed what we meant to say, even though the PDFs closely follow the meridional behavior of the mean. We will better phrase it in the revised manuscript.

Line 232: (see my general marks earlier): Why can the positive and negative extremes overlap? In my simple-minded thinking, the extremes of a PDF represent the opposite tails of the PDF, so I do not understand why and how these can "overlap". I probably did not understand your definition of "extreme", but it may help other readers if you could say here why this is so.

We agree that this needs to be clarified in the revised manuscript - see our reply to the comment above.

Line 233: What do you mean here by "pattern"?

We acknowledge that the word "patterns" might be confusing in this context, and we will replace it by "features".

Line 268: shouldn't it be "…. higher zonal variability in the former…."?!

If the reviewer refers to line 286, that is correct. We will switch the order of JJA and DJF at the beginning of the sentence.

Line 311 (and similar at some other line): you talk about a "midlatitude channel", but this term is misleading as it should be reserved for a geometric setup with walls at the southern and northern boundary of the channel. As far as I can tell, you are dealing with spherical geometry, never with true "channel geometry".

We will replace the term "channel" with a less specific notion. We would however like to observe that a cylindrical geometry is indeed taken into account for the Fourier decomposition, as its symmetry is required in order to obtain the different wavenumber modes of the transport.

Line 318: a heatwave cannot possibly be a "case study". You probably mean that this heat wave is a "case".

Agreed. We will remove the word "study".

Line 345: what are "higher-scale eddies"? I would prefer the term "smaller scales".

Agreed. We will edit the text accordingly.

Line 385 ff: (see my earlier remarks): Do your results really suggest that the modes associated with heat flux extremes are hemispheric in scale? It seems to me that this is a necessary consequence of your methodology that focuses on individual wavenumbers. If so, it cannot possibly be a result of your study.

We will rephrase the sentence in order to state more clearly what we mean, as per our reply to the aforementioned general comment.

Typos etc.

Line 34: must be ".... a poleward transport...."

Fig 9 and 10: letters a and b missing to denote the two different panels.

These typos and missing labels will be rectified in the revised manuscript.

---

## Author Comment (AC2)

**Reviewer 2**

This study investigates the characteristics of the extreme meridional energy transport associated with various zonal scales, using the reanalysis data. Using Extreme Value Theory, extreme events of the meridional energy transports are identified, and their associated zonal wavenumbers and meteorological patterns are analyzed. They found that extreme energy transports are, in general, associated with planetary (synoptic) scale wave during boreal winter (summer). Further, they connect those extreme energy transport events with commonly known teleconnection patterns. The topic and the results of this paper generally fits the aim of the WCD and would improve the scientific community's knowledge about the meridional energy transport.

However, I found that the manuscript's writing and the scientific results are vague. I think the Introduction needs more strong motivation and hypothesis, and the Methodology section should be written with more details as readers with meteorological background might not be familiar with advanced statistical method such as EVT. More importantly, I found it very difficult to digest the meteorological and dynamical interpretations of the extreme events presented in the Result and Discussion sections. My specific comments are presented below.

We wish to thank the reviewer for the timely and accurate revision, for the constructive criticisms that have inspired fruitful discussions on how to improve the conveyance of the main message and better placement in the context of current research on the role of extremes and energy exchanges in a changing climate. Replies to specific comments are marked below in red, illustrating the changes that will be proposed in a revised version of the manuscript.

**Introduction**

First three paragraphs introduce general information of the meridional energy transport, and L48-51 only mentions the plan of this paper. Yet, I think the introduction can be improved by adding more motivations and hypothesis. Here are some suggestions.

• Why do we need to pay attention to the energy transport extremes at different length scales? I think L33-35 touches this issue, but it is not so clear to me how planetary waves can oppose the total transport. I think it just depends on the structure and the phase of the wave itself, and thus one cannot make a general statement about it. Can you provide some more references or more explanations?

A substantial body of literature is pointing towards the role of meridional energy transports in communicating the climate change signal towards the high latitudes, especially when one takes into account the latent energy part of the moist static energy (Hwang et al. 2011, Skific and Francis, 2013). More recently, it has been found that transient eddies are mostly responsible for the convergence of atmospheric latent energy towards the high latitudes (Boisvert and Stroeve, 2015). The intermittent and sporadic nature of eddy-driven meridional energy transports, as discussed in Woods et al. 2013, Messori et al. 2013; Messori et al. 2015, justifies the importance of detecting, and characterizing in terms of dynamical mechanisms, extreme meridional energy transport events, as they can contribute substantially to the warming of the Arctic (e.g. Rydsaa et al., 2021). A wavenumber, rather than the traditional stationary-transient (Peixoto and Oort, 1992) decomposition, allows an in-depth consideration of such dynamical aspects. For instance, Graversen and Burtu 2016 found that the planetary scales were themselves mostly responsible for the temperature warming in the Arctic

caused by latent energy convergence. Recent works, using the Fourier decomposition introduced in Graversen and Burtu, 2016 or similar methods (cfr. Heiskaanen et al. 2020), found that planetary scales in atmospheric energy transports are substantially important for the Arctic amplification, unlike synoptic scales, and that their contribution has significantly increased in the last decades (cfr. Rydsaa et al. 2021). Despite the work looking into extreme events in order to understand Arctic changes, and hinting at the role of weather systems (e.g. Liu and Barnes 2015) and storm tracks (Dufour et al. 2016), to our best knowledge no effort has been made yet to link dynamical configurations of the atmosphere to extremes in the meridional energy transports. This is the scope of our analysis and we will rearrange the Introduction accordingly, accounting to the brief excursus that has been summarized above.

Regarding the peculiarity of planetary-scale transports, sometimes opposing the total transports, we believe that the sentence at II. 33-35 has to be rephrased, as it may wrongly suggest that the planetary-scale transport is not a component of the overall total transport. What we aimed at observing here, was that the contribution of planetary scales to the overall transport, especially in the weak JJA transports, as found in Lembo et al. 2019, can transport energy equatorward, rather than poleward, in this way opposing the sign of the total transport. This is a signature of a counter-gradient eddy transport, that goes against the usual baroclinic conversion observed in mid-latitudes, and points towards a different mechanism that is yet to be understood. Understanding the role of planetary scales, as also pointed out in Rydsaa et al. 2021, is increasingly relevant in a warming climate, with ongoing debate on whether the decreasing meridional temperature gradient by Arctic amplification is setting more favorable conditions for the propagation and growth of planetary-scale waves (cfr. Barnes and Polvani 2013; Fabiano et al. 2021; Moon et al. 2022).

• What is the main hypothesis? What do authors expect to find out by analyzing the different component of the meridional transport, for different seasons?

Regarding the main hypothesis of this work, see our comment above. Regarding the relevance of breaking down the meridional transports and their components to different seasons, on one hand it has been found that the role of atmospheric energy transport is particularly relevant in winter, on the other hand not as much interest has been put into the development of waves and the strength of the meridional heat transports in summer. We already found in Lembo et al. 2019 that the relative contribution of synoptic and planetary scales radically changes in the two seasons, especially because of the planetary scales. In this work, it is our intention to develop further such ideas, retaining the focus on extreme events during the two seasons.

Method:

• L92: Authors have defined the planetary scale to be k=1 to 5, while some previous researches have defined waves with zonal wave number 1 to 3 as planetary scale waves and wavenumber 4 or higher as synoptic scale waves (cf. Baggett and Lee 2015; Shaw 2014 https://doi.org/10.1175/JAS-D-13-0137.1). Therefore, some discussion to justify the author's choice of the threshold between planetary and synoptic scale wave number would be helpful. Also, in L276, authors refer k=5 as a synoptic scale wave which is not consistent with the definition of the synoptic scale used in this paper.

**Correct. We will change I. 276 accordingly.**

As for the choice of the wavenumber ranges, we acknowledge that the choice of the threshold for the separation between planetary and synoptic scales is somehow arbitrary and deserves some more

justification, that will be provided in Section 2.2.1. We hereby note that our approach basically follows the one of Graversen and Burtu, 2016 (and subsequently Lembo et al. 2019). The threshold results by the consideration that different length scales correspond to the same wavenumber at different latitudes (as discussed in Heiskanen et al. 2020), our choice being consistent with lower threshold (k=3 for Rydsaa et al. 2021, k=4 for Shaw et al. 2014) at higher latitudes. We argue that the choice of the wavenumber groupings does not substantially affect the interpretation of our results.

Nevertheless, Figure R1 shows what a different grouping, for instance consistent with Rydsaa et al. 2021 would imply for our interpretation of the extreme events. The different grouping consists here of the zonal wavenumbers, k=1-3, that can be denoted as "ultra-long planetary waves", k=4-6, as "planetary waves", k=7-9 as "synoptic waves". The panel k=0, i.e. zonal mean, is left unchanged. Starting from the DJF season (left panel), it is confirmed that ultra-long planetary waves are dominant, especially in the definition of "poleward" extremes at higher latitudes, whereas other planetary waves are relevant at all latitudes (with homogeneous contribution across latitudes). The contribution of synoptic waves is weaker, although comparable to planetary waves, especially in the equatorward half of the mid-latitudinal channel. Looking at JJA, the three eddy contributions are comparable, with planetary and synoptic waves contributing at lower and higher latitudes. Interestingly, ultra-long and planetary waves have a significant part of their PDFs related to equatorward extremes in the negative domain. In other words, we claim that both components transport energy "counter-gradient", as opposed to the total transport.

Overall, one might notice that:

- synoptic-scale waves defined in this way are remarkably homogeneous in latitudes and constant across seasons, so that the only appreciable change is in the position of the peak. This is somehow coherent with our approach, considering k=6-10 as the synoptic wave domain;
- ultra-long planetary waves play a dominant role in shaping the extremes in the DJF season, and this is consistent with the "fine tuning" of the spectrum that we perform at Figures 7 and 8. Not differently, JJA extremes are characterized by the coexistence of comparable planetary and synoptic contributions, although the former ones still dominate poleward transports, while ultra-long waves hardly distinguish between poleward and equatorward extremes;
- the regrouping of wavenumbers allows us to observe that the strength of the extremes is in all cases dependent on the shape of the median meridional section. The k=1-5 grouping showing non correlation with the median, is the result of the latitudinally homogeneous median in the k=4-6 range, plus the weaker contribution by ultra-long waves in low latitudes;

We will consider including these arguments in a dedicated Appendix.

**Figure R1:** PDFs of total (filled contours) and extreme poleward (red contours) and equatorward (blue contours) DJF (left) and JJA (right) meridional energy transports over the 1979-2012 period in ERA5. (a) Sum of all wavenumber contributions; (b) k=0 (zonal mean); (c) k=1-3 (ultra-long planetary scales); (d) k=4-6 (planetary scales); (e) k=7-9 (synoptic scales). Total PDFs have been normalized at each latitude; PDFs of extremes are weighted at each latitude by the number of extreme events. Yellow lines denote mean values.

L124-126: I think this is a serious issue. If authors decided to remove the trend, then they should remove it from the entire grid point. Removing trend only at certain latitudinal band may result a physical unrealistic field and further analysis based on these data would make the readers to suspect the results. So, I suggest either do not remove the trend or remove the trend from the entire grid point. Or at least, authors should provide some information (perhaps as a supplementary figures) that qualitative results don't change regardless of the de-trending method (Even if the results may qualitatively remain same, authors would need to justify their choice anyway).

Agreed. As this was also brought up by reviewer 1, we show in Figure R2 how the thresholds look like if the trend and seasonal cycle are removed everywhere. Whereas the detrending alone (not shown) does not noticeably change the results, the deseasonalizing has a slight effect in DJF, as it can be seen in the first two rows of the figure. There is (left) a small discontinuity at latitude 45°, coinciding with the latitude north of which no deseasonalization has been originally performed. This small discontinuity disappears when the transports are deseasonalized at each latitude (right). However, the change is really minor, especially for the selected thresholds marked by the blue dots, thus it does not affect our results. In case of JJA, there is no noticeable change between the two procedures (left vs. right).

---

## Author Response (AR1)

**Reviewer 1**

This paper presents a systematic analysis of extremes in the zonally averaged meridional heat transport and how they are related to weather regimes and preferred zonal wavenumbers. The work is based on several decades of reanalysis data and draws on the results from earlier publications. Overall, the authors argue that their results are consistent with previous results regarding weather regimes, dominant wavenumbers, and how they are related to heat transport extremes. The current analysis makes explicit the role of planetary versus synoptic scales in this context. The question of extreme events is of primary importance in our science, and a detailed analysis such as the present one is welcome. I can see this as a publication in WCD.

Yet, I have a few issues. To be sure, I need to say that I am not an expert in the present topic, rather I consider myself as representative of a typical reader of WCD. As such, I had a hard time in several of the more technical sections to understand what the authors have really done. This is probably due to the fact that the text seems to be primarily directed at the expert, who is familiar with a string of earlier publications from the same group. I have no doubt that the analysis is performed in a proper way; however, I suspect that this is hard to appreciate by the average reader of WCD.

As a way out I suggest that the authors should make a serious attempt to more pedagogically introduce the concepts used in their analysis as well as in their results sections. Instead of just providing the references to multiple generations of previous publications, assuming that each reader is familiar with those papers, the authors should add some advice to the not-so-expert reader trying to introduce and/or summarize these earlier developments on a conceptual level. This would increase the readability of and add great value to the paper.

In addition, the paper would benefit if the authors could add some non-technical guidance to the reader as to what these results mean in more meteorological terms and what the implications are. To be sure, you draw a few interesting conclusions. However, you should make a more serious attempt to connect these conclusions to the more technical parts of the paper. Again, I do not doubt the validity of the results or the conclusions; I just feel that this paper would make a much stronger impact if such meteorological guidance were available and if the technical and the interpretatory parts of the paper are connected in a more seamless fashion. Also, you often point out the consistency with earlier results, and by doing so some readers may get lost and left unclear about what is really new about this paper; therefore, it would be good you could point out more explicitly what is new in the current paper.

We wish to thank the reviewer for their careful assessment of our work, insightful comments and constructive criticisms. We acknowledge that the technical description of the methodology is somewhat difficult to follow for a non-expert reader. In the revised manuscript, we tried to better address the techniques in a more pedagogic way and focused on an insightful explanation of the rationale behind the choice of the methodology, mainly in the Introduction and Discussion sections. A careful assessment of the novelty of our results, and a comparison with previous results is also a task of the revised version of the manuscript.

Examples

Let me provide a few examples illustrating the major issue made above. As I said, some work for improvement would be appreciated in the interest of a broader readership.

For instance, equations (3) and (4) were unclear to me at my first reading. If you do a Fourier decomposition of a field and multiply two such fields (as you have to do to compute a heat flux), you obtain a double sum, one for each expansion. You can, then, sort this double sum according to the resulting zonal wavenumber, and this results in each Fourier coefficient of the heat flux being a sum of many terms from the individual terms (v and E) that just happen to add up to the zonal wavenumber in consideration. This is what I would have expected in equations (3) and (4), but your method is different.

To be sure, I could have read the quoted papers in order to educate myself (to be honest, a cursory look into Graversen and Burtu 2016 did not help me a lot), but I would not be too optimistic regarding the readiness of the average WCD reader to do so. Instead, I would have appreciated not just a short "summary" of those earlier methodological developments, but rather a conceptual introduction on a somewhat higher "meta-level".

In the end, the point here is that you consider zonally integrated fluxes, and Parseval's theorem allows one to express the zonal integral of a quadratic quantity as a single sum over all wavenumbers like in (4). The other important point here is that the sum of all individual components such as (3) and (4) is equal to the total, zonally integrated heat flux, which you refer to as "wavenumber decomposition" later in your text. Implicitly, you heavily draw on this property in the rest of the paper. A corresponding hint in the method's section would have helped me a lot!

**Reply:** The reviewer correctly points out that the Fourier decomposition method has limitations, that our analysis is constrained by consideration of zonally integrated fluxes, and that this has not been sufficiently brought up, neither in the introduction, nor in the Methods section. This  caveat explains why the transports have to be interpreted in hemispheric budgetary terms. Further, as the extreme detection, being zonally integrated, does not give information on localized features of the dynamics, we start our analysis from weather regimes identified in several regions. This is complemented by looking at the composite means of geopotential anomalies.

**Changes:** the introduction has been restructured, better framing the context of our wavenumber decomposition. Particularly, ll. 22-27 describe the usual transient-stationary decomposition while ll. 29-49 introduce the wavenumber decomposition of meridional energy transports and how these have been used to observe aspects of the atmospheric circulation, making explicit the constraints of the zonally integrated approach. This has been brought up again in some parts of the Discussion (ll. 421-422, ll. 438-440) and Conclusion (ll. 473-475) sections. Regarding the methodology, we referred to the Parseval's theorem at ll. 105-106.

To provide a second example, in Fig. 4b it was not clear to me at first why the extremes do not just represent the tails of the distribution from the color fill (just like in a box-and-whisker plot). This is what I would have expected initially. The same problem arises in the text on line 232: how can possibly the "equatorward and poleward extremes largely overlap"? Shouldn't the extremes represent opposing ends of a PDF? If so, it is hard to see how they can overlap. The solution to this problem probably depends on how you defined the extremes and their PDF: the extremes are defined without reference to a wavenumber, and this implies that the existence of an extreme does not have to be reflected in the PDF of each and every wavenumber. Is that right? Other readers may have a similar problem, and some explanation would be very helpful. In addition, reading this (and related) plots is made more difficult due to the fact that the caption does not give contour intervals for the dashed isolines.

**Reply:** We agree with the Reviewer that it is not clear from the current text how the meridional energy transport extremes have been selected. This is done on the population of total heat transports. As a consequence, the PDFs of the wavenumber contributions to the extremes in the two tails of the distribution can overlap.

**Changes:** ll. 287-288 refers to the selection of extreme events from zonally averaged total transports. Contour labels have been added to the extreme transport PDFs in Figures 3 and 4.

In the last section, you draw some interesting conclusions, which I was not always able to relate to the core of your analysis. For instance, you say that "planetary scales determine the strength and meridional position of the synoptic-scale baroclinic activity with their phase and amplitude": where exactly have you shown this? How can you make statements about the wave's phase, which (as far as understand) is unavailable from just looking at the zonally integrated heat transport? Similar reservation I have with the conclusions on lines 371-373. I feel that you need to tell the reader somewhat more explicitly how you arrive at these conclusions and which part of the analysis your conclusion is based on.

**Reply:** The zonally integrated approach does not allow to explicitly address the phase of the waves, as correctly pointed out by the reviewer. However, the combined view of dominant wavenumbers, weather regimes and, to some extent, composite analysis, allows us to infer qualitatively how scales interact when total meridional heat transport extremes occur. We argued, in particular, that DJF extremes are the result of a planetary-scale modulation of synoptic-scale eddies, given that:

- the planetary-scale component is the dominant feature of DJF transports, especially north of 40N, with the two tails of the extremes well separated (Figure 3) and previous evidence (Lembo et al. 2019) that synoptic-scale and planetary-scale component extremes are rarely co-located;
- k=2-3 are the dominant zonal wavenumbers for the transport anomalies, as shown in Figure 7, no matter what the sign of the extreme is;
- as Figure 5 shows, poleward (equatorward) extremes are denoted by increased (decreased) frequency in NAO-/AO/PT regimes and decreased (increased) frequency of NAO+/ALR regimes. This is also partly resembled by Figure 10, where z500 composite means are shown;

Based on this we qualitatively argue that planetary scales modulate synoptic-scale baroclinic activity in the population of extremes, with the largest changes in weather regimes related to increased/decreased blocking frequencies and such ultra-long planetary-scale waves dominating the heat transport in the population of extreme heat transports.

**Changes:** we removed explicit reference to the phase of planetary waves in the Discussion section. We also added a sentence at ll. 442-444 to explain how information on phases of the dominant waves can be inferred from jointly considering weather regime frequency differences and composite maps.

Take another example: you say on line 385 that "...our results emphasize that the modes related to energy transport extremes are hemispheric in scale". What part of your analysis is this statement based on? My point here is that the chief instrument in your analysis is the investigation of the zonally integrated heat flux, and this leads (almost by design) to "modes" that can be expected to be hemispheric in scale rather than very local or small-scale. In summary, all of these conclusions may be

well justified, it was just not easily visible for me. The authors should make an attempt for improvements in this direction.

**Reply:** As above, we acknowledge that the zonally integrated approach somehow limits the opportunity to observe regional-scale features of the meridional heat transport extremes across the NH mid-latitudes. The rationale behind looking at weather regimes in relation to these occurrences was that there could be in principle some mid-latitude regions where the atmospheric circulation is more sensitive to excited meridional heat transport. Looking at the changing occurrences of preferred weather regimes, together with dominant zonal wavenumbers, we found that this is probably not the case, and that the peculiar role of planetary scales may influence several regions at the same time, as also suggested by the consideration of the 2010 Russian heat wave case. Once again, the significance of composite means (see reply to reviewer 2) complements this. In the revised text, we will clarify the logical reasoning that brought us to draw the conclusion stated on l. 385, and the fact that it is based on a qualitative evaluation of our results. This conclusion is consistent with the referenced works by Comou, Petoukhov, Kornhuber on quasi-resonant amplification (QRA), and we hope that we can expand on the dynamical linkages between co-recurrent blockings and meridional heat transports in a future work.

**Changes:** the example of the 2010 Russian heat wave is better framed in the context of the results provided at ll. 391-405. We emphasized at ll. 471-475 in the Conclusions section that our approach allowed to show how zonally integrated transport extremes reflect on regional features of the atmospheric circulation, despite there was no reason to believe that this would be the case, if the population of extreme events would encompass different phases of the dominant waves. At ll. 403-405 and 487-489 we stressed that this has a lot to do with the existence of recurrent patterns of the general circulation and the possibility to study the dynamical system through the lens of the persistence property.

Minor comments

Line 16: This is somewhat advanced material for the start of an introduction. Presumably you talk about vertically averaged moist static energy, right? In the tropics the vertical change of moist static energy is close to zero, because the increase of potential temperature with altitude is, to a large extent, compensated by a decrease of water vapor mixing ratio.

**Reply:** Agreed. We believe it is sufficient, in this context, to refer to it as simply "heat".

**Changes:** we replaced "moist static energy" by "heat" at l. 16.

Line 125: you remove the linear trend only in certain latitude bands. Why does this not create awkward discontinuities at the boundaries of these ranges?

**Reply:** The conditional removal of the linear trend was accomplished with the sole purpose of a rigorous application of the EVT-based selection algorithm, particularly to the convergence analysis of the percentile threshold. Thus, it was not applied to the subsequent analysis.

As this was also brought up by reviewer 2, we show in Figure R2 how the thresholds look like if the trend and seasonal cycle are removed everywhere. Whereas the detrending alone (not shown) does not noticeably change the results, the deseasonalizing has a slight effect in DJF, as it can be seen in

the first two rows of the figure. There is (left) a small discontinuity at latitude 45°, coinciding with the latitude north of which no deseasonalization has been originally performed. This small discontinuity disappears when the transports are deseasonalized at each latitude (right). However, the change is really minor, especially for the selected thresholds marked by the blue dots, thus it does not affect our results. In case of JJA, there is no noticeable change between the two procedures (left vs. right).

[Figure]

**Figure R1:** Meridional section of threshold values for meridional energy transport extremes selection considering different percentiles (the selected threshold is highlighted in blue, as in Figure 1 of the manuscript). In the left column, transports have been deasonalised and detrended only where necessary, in the right column everywhere: (1st row) DJF, poleward, (2nd row) DJF, equatorward, (3rd row) JJA, poleward, (4th row) JJA, equatorward.

**Changes:** at ll. 154-158 we mentioned the sensitivity test described above.

I suggest to increase the size of the panels in Fig. 1 and 2.

**Reply:** Agreed.

**Changes:** we changed the layout of Figure 1, in order to have 2 columns instead of 3 (the panel about the scale parameter has been removed, as it was not discussed anywhere). Figure 2 has also been changed in order to have less empty space.

Panel 1c, y-axis-label: the threshold should have dimensions, right?! How about the physical dimensions of the scale and the shape parameter?

**Reply:** The shape parameter is a non-dimensional parameter and the scale parameter has the dimension 10^15 W.

**Changes:** we removed the scale parameter panel.

Fig 3 and 4: How did you normalize the PDFs? It seems to me that integrating by eye over the heat transport at a fixed latitude one may obtain values larger than 1. Or put the other way: what units does the plotted PDF have? Is it really (10^15 W)^(-1)? How should I read the red and blue dashed contours corresponding to the extreme situations (no contour interval given....).

**Reply:** PDFs are normalized by the maximum value across all latitudes, in order to compare the different ranges at different latitudes. In order to account for the different number of extremes at different latitudes, the Friedman-Diaconis rule (Friedman and Diaconis, 1981) is first applied to determine the correct number of bin elements for the discretized PDF, then the kernel smoothing estimate of the PDF (Bowman and Azzalini 1997) is computed. A preliminary condition is applied on the number of bins, that has to be at least 2. PDFs are then normalized at each latitude by their maximum value, so that their maximum value is 1. For graphical purposes, the obtained PDFs for the extremes are finally interpolated on the same number of bins as the filled contour plots of the overall population, which is the same at all latitudes.

**Changes:** we changed Figures 3 and 4 in order to have the same x-axis range. We changed the normalization in order to have the same normalization at all latitudes, so that the information about different ranges and shapes at different latitudes is included. The isolines for the extreme transport PDFs are provided with labels. The figure caption in Figure 3 now includes details on how the PDFs have been obtained and how they are shown.

Line 217: "…. the PDF steeply decays towards the high latitudes….", I understand what you want to say, yet, it is not really well expressed. You probably want to say that the mean or median of the PDF decreases as one goes to higher latitudes.

**Reply:** That is indeed what we meant to say, even though the PDFs closely follow the meridional behavior of the mean. We will better phrase it in the revised manuscript.

**Changes:** we changed ll. 259-260 as suggested by the reviewer.

Line 232: (see my general marks earlier): Why can the positive and negative extremes overlap? In my simple-minded thinking, the extremes of a PDF represent the opposite tails of the PDF, so I do not understand why and how these can "overlap". I probably did not understand your definition of "extreme", but it may help other readers if you could say here why this is so.

**Reply:** We agree that this needed to be clarified in the revised manuscript.

**Changes:** at ll. 267-268 we tried to better explain how the extreme transports have been selected.

Line 233: What do you mean here by "pattern"?

**Reply:** We acknowledge that the word "patterns" might be confusing in this context.

**Changes:** we changed the word "patterns" to "features" as suggested by the reviewer.

Line 268: shouldn't it be "…. higher zonal variability in the former…."?!

**Reply:** If the reviewer refers to line 286, that is correct.

**Changes:** we changed l. 359 accordingly, switching the order of JJA and DJF at the beginning of the sentence.

Line 311 (and similar at some other line): you talk about a "midlatitude channel", but this term is misleading as it should be reserved for a geometric setup with walls at the southern and northern boundary of the channel. As far as I can tell, you are dealing with spherical geometry, never with true "channel geometry".

**Reply:** We will replace the term "channel" with a less specific notion. We would however like to observe that a cylindrical geometry is indeed taken into account for the Fourier decomposition, as its symmetry is required in order to obtain the different wavenumber modes of the transport.

**Changes:** we replaced the word "channel" with "band" at several occurrences throughout the manuscript.

Line 318: a heatwave cannot possibly be a "case study". You probably mean that this heat wave is a "case".

**Reply:** Agreed

**Changes:** we removed the word "study" at l. 391.

Line 345: what are "higher-scale eddies"? I would prefer the term "smaller scales".

**Reply:** Agreed.

**Changes:** We edited the text accordingly at l. 425.

Line 385 ff: (see my earlier remarks): Do your results really suggest that the modes associated with heat flux extremes are hemispheric in scale? It seems to me that this is a necessary consequence of your methodology that focuses on individual wavenumbers. If so, it cannot possibly be a result of your study.

**Reply:** see our comment above.

**Changes:** We added two new sentences at ll. 471-475 in the Conclusions section.

Typos etc.

Line 34: must be "…. a poleward transport…."

Fig 9 and 10: letters a and b missing to denote the two different panels.

**Reply:** agreed.

**Changes:** all typos and missing labels have been corrected in the new version of the manuscript.

**Reviewer 2**

This study investigates the characteristics of the extreme meridional energy transport associated with various zonal scales, using the reanalysis data. Using Extreme Value Theory, extreme events of the meridional energy transports are identified, and their associated zonal wavenumbers and meteorological patterns are analyzed. They found that extreme energy transports are, in general, associated with planetary (synoptic) scale wave during boreal winter (summer). Further, they connect those extreme energy transport events with commonly known teleconnection patterns. The topic and the results of this paper generally fits the aim of the WCD and would improve the scientific community's knowledge about the meridional energy transport.

However, I found that the manuscript's writing and the scientific results are vague. I think the Introduction needs more strong motivation and hypothesis, and the Methodology section should be written with more details as readers with meteorological background might not be familiar with advanced statistical method such as EVT. More importantly, I found it very difficult to digest the meteorological and dynamical interpretations of the extreme events presented in the Result and Discussion sections. My specific comments are presented below.

We wish to thank the reviewer for the timely and accurate revision, for the constructive criticisms that have inspired fruitful discussions on how to improve the conveyance of the main message and better placement in the context of current research on the role of extremes and energy exchanges in a changing climate. Replies to specific comments are marked below in red, illustrating the changes that will be proposed in a revised version of the manuscript.

Introduction

First three paragraphs introduce general information of the meridional energy transport, and L48-51 only mentions the plan of this paper. Yet, I think the introduction can be improved by adding more motivations and hypothesis. Here are some suggestions.

- Why do we need to pay attention to the energy transport extremes at different length scales? I think L33-35 touches this issue, but it is not so clear to me how planetary waves can oppose the total transport. I think it just depends on the structure and the phase of the wave itself, and thus one cannot make a general statement about it. Can you provide some more references or more explanations?

**Reply:** A substantial body of literature is pointing towards the role of meridional energy transports in communicating the climate change signal towards the high latitudes, especially when one takes into account the latent energy part of the moist static energy (e.g. Hwang et al. 2011, Skific and Francis, 2013). More recently, it has been found that transient eddies are mostly responsible for the convergence of atmospheric latent energy towards the high latitudes (Boisvert and Stroeve, 2015). The intermittent and sporadic nature of eddy-driven meridional energy transports, as discussed in Woods et al. 2013, Messori et al. 2013; Messori et al. 2015, justifies the importance of detecting, and characterizing in terms of dynamical mechanisms, extreme meridional energy transport events, as they can contribute substantially to the warming of the Arctic (e.g. Rydsaa et al., 2021). A wavenumber, rather than the traditional stationary-transient (Peixoto and Oort, 1992) decomposition, allows an in-depth consideration of such dynamical aspects. For instance, Graversen and Burtu 2016

found that the planetary scales were themselves mostly responsible for the temperature warming in the Arctic caused by latent energy convergence. Recent works, using the Fourier decomposition introduced in Graversen and Burtu, 2016 or similar methods (cfr. Heiskaanen et al. 2020), found that planetary scales in atmospheric energy transports are substantially important for the Arctic amplification, unlike synoptic scales, and that their contribution has significantly increased in the last decades (cfr. Rydsaa et al. 2021). Despite the work looking into extreme events in order to understand Arctic changes, and hinting at the role of weather systems (e.g. Liu and Barnes 2015) and storm tracks (Dufour et al. 2016), to our best knowledge no effort has been made yet to link dynamical configurations of the atmosphere to extremes in the meridional energy transports.

**Changes:** Above mentioned considerations have been taken into account to rearrange the introduction, in particular ll. 29-53. Regarding the peculiarity of planetary-scale transports, sometimes opposing the total transports, the highlighted sentence has been rephrased at ll. 42-44. In general, we aimed in the ll. 41-49 paragraph at emphasizing that all waves are components of the total transport, upon which the extremes are selected. The planetary-scale contribution, as found in Lembo et al. 2019, can be negative (meaning "equatorward") for exceptionally weak JJA transports. In a quasi-geostrophic framework, the thermodynamic interpretation of such counter-gradient transport that goes against the usual baroclinic conversion is yet to be understood, and justifies looking into the modes of atmospheric variability, in order to investigate the mechanisms behind it, as mentioned at ll. 49-51.

- What is the main hypothesis? What do authors expect to find out by analyzing the different component of the meridional transport, for different seasons?

**Reply:** Regarding the main hypothesis of this work, see our comment above. Regarding the relevance of breaking down the meridional transports and their components to different seasons, on one hand it has been found that the role of atmospheric energy transport is particularly relevant in winter, on the other hand not as much interest has been put into the development of waves and the strength of the meridional heat transports in summer. We already found in Lembo et al. 2019 that the relative contribution of synoptic and planetary scales radically changes in the two seasons, especially because of the planetary scales.

**Changes:** we have deeply restructured the Introduction section, on one hand emphasizing the difference between the usual transient-stationary decomposition and the wavenumber/wavelength decomposition here adopted, on the other hand stating more clearly why this latter approach is particularly suitable for drawing comparisons between (zonally integrated) meridional energy transport and modes of atmospheric variability.

Method:

- L92: Authors have defined the planetary scale to be k=1 to 5, while some previous researches have defined waves with zonal wave number 1 to 3 as planetary scale waves and wavenumber 4 or higher as synoptic scale waves (cf. Baggett and Lee 2015; Shaw 2014 https://doi.org/10.1175/JAS-D-13-0137.1). Therefore, some discussion to justify the author's choice of the threshold between planetary and synoptic scale wave number would be helpful. Also, in L276, authors refer k=5 as a synoptic scale wave which is not consistent with the definition of the synoptic scale used in this paper.

**Reply:** As for the choice of the wavenumber ranges, we acknowledge that the choice of the threshold for the separation between planetary and synoptic scales is somehow arbitrary and deserves some

more justification, that will be provided in Section 2.2.1. We hereby note that our approach basically follows the one of Graversen and Burtu, 2016 (and subsequently Lembo et al. 2019). The threshold results by the consideration that different length scales correspond to the same wavenumber at different latitudes (as discussed in Heiskanen et al. 2020), our choice being consistent with lower threshold (k=3 for Rydsaa et al. 2021, k=4 for Shaw et al. 2014) at higher latitudes. We argue that the choice of the wavenumber groupings does not substantially affect the interpretation of our results. Nevertheless, Figure R1 shows what a different grouping, for instance consistent with Rydsaa et al. 2021 would imply for our interpretation of the extreme events. The different grouping consists here of the zonal wavenumbers, k=1-3, that can be denoted as "ultra-long planetary waves", k=4-6, as "planetary waves", k=7-9 as "synoptic waves". The panel k=0, i.e. zonal mean, is left unchanged. Starting from the DJF season (left panel), it is confirmed that ultra-long planetary waves are dominant, especially in the definition of "poleward" extremes at higher latitudes, whereas other planetary waves are relevant at all latitudes (with homogeneous contribution across latitudes). The contribution of synoptic waves is weaker, although comparable to planetary waves, especially in the equatorward half of the mid-latitudinal channel. Looking at JJA, the three eddy contributions are comparable, with planetary and synoptic waves mostly contributing to poleward extremes in the middle of the channel, and ultra-long planetary waves contributing at lower and higher latitudes. Interestingly, ultra-long and planetary waves have a significant part of their PDFs related to equatorward extremes in the negative domain. In other words, we claim that both components transport energy "counter-gradient", as opposed to the total transport.

Overall, one might notice that:

- synoptic-scale waves defined in this way are remarkably homogeneous in latitudes and constant across seasons, so that the only appreciable change is in the position of the peak. This is somehow coherent with our approach, considering k=6-10 as the synoptic wave domain;
- ultra-long planetary waves play a dominant role in shaping the extremes in the DJF season, and this is consistent with the "fine tuning" of the spectrum that we perform at Figures 7 and 8. Not differently, JJA extremes are characterized by the coexistence of comparable planetary and synoptic contributions, although the former ones still dominate poleward transports, while ultra-long waves hardly distinguish between poleward and equatorward extremes;
- the regrouping of wavenumbers allows us to observe that the strength of the extremes is in all cases dependent on the shape of the median meridional section. The k=1-5 grouping showing non correlation with the median, is the result of the latitudinally homogeneous median in the k=4-6 range, plus the weaker contribution by ultra-long waves in low latitudes;

[Figure]

**Figure R1:** PDFs of total (filled contours) and extreme poleward (red contours) and equatorward (blue contours) DJF (top) and JJA (bottom) meridional energy transports over the 1979-2012 period in ERA5. (a) Sum of all wavenumber contributions; (b) k=0 (zonal mean); (c) k=1-3 (ultra-long planetary scales); (d) k=4-6 (planetary scales); (e) k=7-9 (synoptic scales). Total PDFs have been normalized at each latitude; PDFs of extremes are weighted at each latitude by the number of extreme events. Yellow lines denote mean values.

**Changes:** The choice of the threshold for synoptic-planetary scales distinction is now discussed at ll. 111-116, and the alternative grouping is discussed in the new Appendix C and Figure C1. L. 342 takes into account that the wavenumber k=5 is the largest synoptic scale, according to our choice of the threshold.

- L124-126: I think this is a serious issue. If authors decided to remove the trend, then they should remove it from the entire grid point. Removing trend only at certain latitudinal band may result a physical unrealistic field and further analysis based on these data would make the readers to suspect the results. So, I suggest either do not remove the trend or remove the trend from the entire grid point. Or at least, authors should provide some information (perhaps as a supplementary figures) that qualitative results don't change regardless of the de-trending method (Even if the results may qualitatively remain same, authors would need to justify their choice anyway).

**Reply:** Agreed. As this was also brought up by reviewer 1, we show in Figure R2 how the thresholds look like if the trend and seasonal cycle are removed everywhere. Whereas the detrending alone (not shown) does not noticeably change the results, the deseasonalizing has a slight effect in DJF, as it can be seen in the first two rows of the figure. There is (left) a small discontinuity at latitude 45°, coinciding with the latitude north of which no deseasonalization has been originally performed. This small

discontinuity disappears when the transports are deseasonalized at each latitude (right). However, the change is really minor, especially for the selected thresholds marked by the blue dots, thus it does not affect our results. In case of JJA, there is no noticeable change between the two procedures (left vs. right).

[Figure]

**Figure R2:** Meridional section of threshold values for meridional energy transport extremes selection considering different percentiles (the selected threshold is highlighted in blue, as in Figure 1 of the manuscript). In the left column, transports have been deasonalised and detrended only where necessary, in the right column everywhere: (1st row) DJF, poleward, (2nd row) DJF, equatorward, (3rd row) JJA, poleward, (4th row) JJA, equatorward.

**Changes:** we now discuss the differences in the way the detrending is applied at ll. 154-158.

- L150-157: Authors argue that Figure 1 justifies the choice specific threshold values. However, even after looking at Figure 1, I cannot understand how authors have chose these specific threshold value (ex: 86% percentile for DJF poleward). So more detailed explain regarding this step would be helpful.

**Reply:** The convergence algorithm implies that the shape parameter does not change for thresholds (i.e. percentiles) smaller or larger (depending whether we consider positive or negative extremes) than the chosen one. As the shape parameter converges in different ways at different latitudes (as shown

in Figure 1 for the case of DJF "poleward" extremes) we chose by visual inspection a conservative estimate of the threshold that would account for convergence at all latitudes.

**Changes:** the chosen thresholds are listed at ll. 182-185. The procedure for the choice of the threshold is now better described at ll. 163-171.

- L170: Can you explain why do you first apply EOF analysis before K-means clustering? Can't you just apply K-means clustering to the raw data, or just use the PC timeseries of the first 4 EOFs?

**Reply:** The weather regimes analysis builds on Fabiano et al. (2020), where the details of the procedure are thoroughly discussed. The EOF decomposition is mainly aimed at reducing the dimensionality of the original field, disregarding smaller scales of motion and local noise. This is a common step in most weather regimes identification techniques (see e.g. Dawson et al. 2012, Cassou 2008, Cattiaux et al. 2013, Straus et al. 2007, Dorrington et al., 2022), although we acknowledge that some approaches skip it (Falkena et al., 2020). The reason for cutting at some threshold of variance is filtering out the small scale structures and at the same time having a simpler phase space. Also, the significance of the clustering lowers when considering too many dimensions, and this is generally not desirable.

We performed a sensitivity test on the number of EOFs and the threshold for the variance, and we found that the identified patterns are not significantly changing. As an example, Figure R3a shows for the EAT domain the 4 dominant weather regimes obtained from the first 4 EOFs explaining 55% of the variance, as in Fabiano et al. (2020). Figure R3b, instead, shows the 4 dominant weather regimes obtained when the threshold is raised to 90% and the first 16 EOFs are taken into account, as it is done in our analysis. Differences in regime frequencies for the weather regimes obtained from the two sets of EOFs are generally below 1%, also for the PAC and NH domains (not shown). For these reasons, we can state that the regime patterns are robust to the choice of the variance threshold (Straus et al, 2007; Fabiano et al., 2020), and the limit for a very large number of EOFs clearly has to give the same result as using the full field.

[Figure]

**Figure R3:** Weather regimes of 500hPa geopotential height anomaly (in dam) in the EAT domain obtained with the K-means clustering from (a) 4 EOFs (55% explained variance); (b) 16 EOFs (90% explained variance.

As for the second comment, regarding whether we could just use the PC time series for the analysis, this would represent a viable approach and is briefly addressed in Appendix B. However, we believe that weather regimes have some advantages when discussing the variability of the circulation, starting

from the fact that the regimes represent persistent physical patterns that are effectively realized in the atmospheric circulation, while EOFs need to be "summed up" to reconstitute the physical field. This advantage is also reflected by the fact that weather regime analysis of the mid-latitude circulation is a very active field of research.

**Changes:** ll. 200-204 and ll. 205-209 now reflect the above discussion on the opportunity to reduce dimensionality before K-means clustering is applied, and the information contained in the PC timeseries as compared to what we actually did.

- I question the purpose of finding the weather regimes using a clustering algorithm. It makes more sense to me to directly diagnose the dynamical characteristics of energy transport extremes using the composite map of z500 pattern. My interpretation is that authors are hypothesizing that energy transport extremes should be associated with the identified teleconnection patterns, but that is not necessarily guaranteed. It is possible that each event may have their own circulation structure that may not resemble the known teleconnection patterns. Therefore, some discussion on why authors use clustering algorithm instead of directly diagnosing the circulation composite structures would be helpful.

**Reply:** We thank the reviewer for the substantial comment. We agree that the composite maps of z500 patterns are helpful in analyzing the dominant pattern of the circulation related to the extremes. Indeed, we show such maps in Figures 11 and 12, which confirm the main results of the regime analysis. However, the composites only show the mean field of all extremes, while the information on the variability is averaged out. In fact, taking the composite mean of geopotential height in the selection of extreme event dates, may result in the superposition of events occurring at different longitudes and in different times, leading to severe aliasing of the displayed anomalies. Using weather regimes gives a deeper insight on what kind of circulation patterns are linked to the energy transport extremes, since this allows to recover some information regarding the temporal/spatial variability of such extremes, which is not possible to get from the composite maps only.

A concrete example of how we use the information from composite maps and weather regimes in combination is given in Figure R4. As in Figure 13a-b, we show (a) composite mean maps related to poleward extremes during JF 2010, a period characterized by a significant and persistent cold spell over large swathes of Central Asia, and (b) differences in the frequency of occurrence of weather regimes. We interpret the composite maps as a way to illustrate the mean circulation patterns during the event. It is evident from panel (b) that the regime frequencies analysis is both coherent and more detailed than the composite analysis. In other words, the anomalies associated with these extreme events are distributed in such a way that NAO+/AR, as well as PNA-/ALR, become less frequent, whereas NAO-/AO/PT occurrences are way more frequent. In this particular case, weather regimes' frequencies change in a very similar way to what happens when the overall population of extreme poleward events in all DJF seasons are taken into account (Figures 5a,c,e). Once again, we stress that a verification of the dynamical and meteorological (tele)connections leading to such occurrences of extreme meridional heat transports would require selecting a large database of events to be verified one-by-one, and this goes well beyond the scope of our work.

[Figure]

**Figure R4:** (a) Composite mean maps of z500 anomalies (in dam) for winter (JF) in 2010; (b) Relative variations in absolute frequency of clusters in the population of 2010's JF poleward extremes as a function of the latitude at which extremes are found, for (top) EAT region, (middle) PAC region, (bottom) NH region.

**Changes:** A hypothesis test has been applied on composite means displayed in Figures 11 and 12 (former Figures 9 and 10), whose methodology is described in the caption of Figure 11 and is based on a bootstrapping method. We have better discussed at ll. 354-358 (and at ll. 402-404 and ll. 455-458 for specific cases) how the composite mean maps complement the information about the weather regime frequency differences, dominant wavenumbers and PDFs of the extreme transports

Result

- L221: If the JJA PDF shows positive skewness, are you refereeing more colored contours toward left side of the yellow (mean) line? At least to me, the difference between high and low latitude are not so clear in Figure 4a.

**Reply:** Indeed, we refer to panel a in Figure 4, the equatorward side of the picture. The difference in skewness is determined quantitatively, but we acknowledge that this is not stated clearly enough.

**Changes:** a Figure 5 has been added, with the meridional sections of skewness in the overall population of all events, and for poleward/equatorward events. This is commented at ll. 260-266 for the overall population and at ll. 268-275 for the extremes.

- Can you add some scientific/meteorological interpretations of what it means to have positive skewness, and why positive skewness is an important finding?

**Reply;** As above, in the rearranged section of the text we mention that the positive skewness in the overall PDF relates to the sporadic and intermittent nature of the meridional heat transports, as evidenced in several previos study (Ambaum and Novak, 2014; Novak et al. 2015; Messori et al. 2015; Marcheggiani et al. 2021) and the nonlinear growth of baroclinic eddies in the mid-latitudes (cfr. Novak et al. 2015).

**Changes:** the new Figure 5 and ll. 260-266 and ll. 268-275 currently include a description of skewness as a function of latitude and their implication in terms of the dynamics that they are reflecting..

- L251-253: Can you explain how PT regime (Fig. 2c) can be characterized as lower latitude negative anomalies and high latitude blocking? I think this pattern is rather zonally oriented without a prominent high latitude blocking-like structure or lower latitude signals.

**Reply:** We agree with the reviewer that the sentence as such is unclear and wrongly suggests that the PT regime is denoted by low latitudinal negative anomalies. This was referred to AO and NAO-regimes, we will change the text accordingly. Instead, the PT regime is denoted by high latitudinal troughs and ridges in the PAC region, that are not necessarily attributable to blocking events, but determine large meridional exchanges, that are to some extent relevant for meridional heat transport extremes, as per the rationale of our work.

**Changes:** the text has been rearranged in order to separately consider the Northern Atlantic and Pacific regions. It is now included in ll. 314-317.

- It is somewhat difficult to interpret the results presented in Figs 5 and 6, along with circulation structure presented in Fig. 2. For example, JJA NHC3 is similar to winter AO, and yet they show opposite results in Figs. 5e and 6e. Besides the seasonal difference, can you comment what makes such a difference in the poleward transport even when two circulation fields are dynamically similar? In addition, EATC3 shows increasing frequency in the 30-42°N degree band (Fig. 6b), while its strong circulation patterns are rather located at higher latitude near Greenland and Scandinavia (Fig. 2b). Can you explain how this circulation pattern can be related to the equatorward transport occurring near 30-40°N latitude?

**Reply:** We thank the reviewer for the useful comment. When comparing weather regimes in different seasons, one has to first take into account the different amplitude of the signal. As shown in Figure 2, the scales are different, because the eddy-driven circulation is weaker in summer than in winter. This is a major constraint to make meaningful comparisons about changes in the occurrences of weather regimes over the two seasons. Even if the maps of anomalies are apparently similar, we are looking at two genuinely different aspects of the dynamics, as also hinted at by the different wavenumbers involved (cfr. Figures 3-4, Figures 7-8 and also Figure 3 in Lembo et al. 2019 for a comparison of seasonality in planetary vs. synoptic waves magnitudes seasonality).

Regarding the remote influence of equatorward extremes in JJA for some weather regimes (specifically EATC2 and EATC3), we proposed at ll. 265-266 that this could be related to the centres of baroclinic activity for these events. However, in order to establish a robust connection, an insightful investigation of baroclinic eddy activity should be carried out, so that this interpretation was left out of the main conclusions.

**Changes:** we remarked the differences in weather regimes in the two seasons, even if apparently similar in terms of patterns, at ll. 232-234.

- L286: Authors said '…JJA and DJF differ in the fact that the higher zonal variability in the latter…'. Shouldn't this be the opposite? Figs. 7 and 8 say that JJA is associated with higher zonal variability and higher zonal wavenumber, not DJF.

**Reply:** This is correct, as pointed out by the other reviewer as well.

**Changes:** l. 359 has been changed accordingly.

- L287-288: Authors claim that poleward extremes have more meridionally marked, or zonally uniform, structure compared to the structure of the equatorward extremes. I don't see a clear difference between poleward and equatorward (there are no (a) and (b) in Figs. 9 and 10, so I assume the poleward is the left column and the equatorward is the right column). For example, in Fig.9, both panels of the 45°N-47°N band show zonal wave number 4~5 structure without prominent meridional structure. Also, it is little unclear to me how a relatively zonally uniform circulation structure would favor for a strong meridional energy transport. I would assume meridional wind in a zonally uniform circulation to be small. Providing more detailed reasoning for such an interpretation would be very helpful.

**Reply:** We thank the reviewer for pointing out that the labels were missing. We have revised the figures accordingly. Also, we agree that the claim that the meridional structure is more pronounced for poleward extremes compared to equatorward extremes is not clearly supported by the figures. We would rather point out here that the emergence of patterns consistent with the dominant wavenumbers described in Figure 10 emerges in Figure 11a, with ridges and troughs stronger for poleward than for equatorward extremes (See Figure R5, left). Our interpretation is that it is not the uniformity of the zonal circulation that determines the strength of the transport, but rather the amplitude of the dominant waves.

**Changes:** the sentence at ll. 359-361 better reflects this point of view. Figures 11 and 12 (former 9 and 10) have been corrected.

- Also, Figs. 9 and 10 shows the composite mean of z500 anomalies. Please indicate the sample size of the composite, and significance test of this composite sampled is also necessary.

**Reply:** Indeed, we noticed that Figure 10 was showing incorrect maps at panels 45-47 and 57-60. We apologize for that: the correct figures are provided here.

- **Significance levels:** A bootstrapping-based significance test is performed in order to provide significance to the given maps. The 0-hypothesis that the composite mean value lies within the range of internal variability is tested at the 95-percentile. Maps are shaded where the p-value is larger than 0.05. The left panel shows results for JJA and the right panel for DJF. The new maps shown in Figure R5 are the new Figures 11 and 12.

[Figure]

**Figure R5:** Composite mean of z500 anomalies (in dam) for JJA (left) and DJF (right) extremes. For each season, (a) refers to poleward extremes and (b) to equatorward extremes at 30-33 (top), 45-47 (middle), 57-60 (bottom) given latitudinal bands. The bootstrapping methodology for significance recognition is described above.

- **Size of the composites:** In the following, a small table is provided, regarding the number of samples for each extreme tail and season. We will include that in the revised version of the manuscript.

|  | DJF | | JJA | |
|---|---|---|---|---|
|  | poleward | equatorward | poleward | equatorward |
| 30-33 | 1883 | 1483 | 1214 | 2237 |
| 45-47 | 2052 | 1589 | 1148 | 2260 |
| 57-60 | 2114 | 1501 | 1284 | 2452 |

**Changes:** Figures 11 and 12 now include significance levels. Panels and labels have also been corrected.  The table with the size of the composites is now Table 1 in the revised manuscript.

- Regarding Figs. 9 and 10, the composite of z500 anomalies is helpful to diagnose the circulation structure, but it is yet difficult to tell where the energy transport is prominent. I think that plotting the composite of anomalous *vE* would help readers to diagnose the prime location(s) of the meridional energy transport.

**Reply:** We thank the reviewer for the suggestion. Our Fourier decomposition relies on the fact that zonally averaged fluxes are taken into account. This implies that we lose information on the region where the convergence and divergence of heat occurs. This is at the same time a caveat and part of the rationale behind this work. In fact, the zonally integrated view allows one to consider the problem from a budgetary point of view: in other words, since extreme meridional heat transports can take up to 55% of the energy that is transported poleward across the midlatitudes in a season (cfr. Messori et

al. 2015), which wavenumbers are mostly responsible for such transports? This is one of the questions we aimed to answer with this work. We also realized that a view of the situation in terms of transport ratios allocated in the different wave groups is missing.

To give a sense of what the situation is, when looking at geographical maps of meridional heat transports and why they are not informative because of the zonally integrated approach, in Figure R6 we show the composite time means of meridional heat transports for poleward extremes taken in two subsets of the considered period: (left) JJA in 2010, characterized by the occurrence of the Russian heat wave, as described in the manuscript, and (right) JF 2010, coincident with the Mongolian Dzud cold spell event. For each row, extreme events captured at different latitudes (30-33, 45-47, 57-60 bands) are selected, as in Figures 9 and 10 of the manuscript. Transports are shaded in gray where significant, according to a 1-sigma significance level.

[Figure]

**Figure R6:** Time mean of meridional energy transports for poleward extremes in the 30-33 (top), 45-47 (middle) and 57-60 (bottom) latitudinal bands, selected according to the algorithm described in

the manuscript, for the JJA 2010 (left) and JF 2010 (right) periods. Areas with mean transport exceeding 1-sigma standard deviation are shaded in gray.

It is clear that, even for such a small subset for events (ranging between 40 and 100 for each of the latitudinal band and tail), a direct attribution of extreme events is difficult, and significant regions are relatively geographical constrained, though informative, in some respects. For instance, extreme events for all three bands are characterized by significant positive (poleward) transports in Central and Western North America, denoting a teleconnection pattern that is consistent with the wave 5 pattern evidenced in Kornhuber et al. 2020. Looking at the Dzud event, a poleward transport emerges in the Northern Pacific and Northern Atlantic up to Greenland. That is fairly consistent with a dynamical pattern denoted by negative geopotential anomalies over Siberia and Mongolia, and an equatorward advection of cold air from the Pole affecting large portions of central Asia.

**Changes:** we included a new Figure 6, showing the ratios of the different contributions over the total transport, as a function of latitude, for all events and poleward/equatorward extremes. This is commented at ll. 294-305. The relevance and constraints of the choice to work with zonally integrated transports is now discussed in some parts of the Discussion (ll. 421-422, ll. 438-440) and Conclusion (ll. 473-475) sections.

Discussion

Comments on QRA and heatwaves:

L302-328: Authors argue that the heat waves are related to the poleward energy transport and present the year 2010 as an example of the extreme poleward energy transport. I found this interpretation is somewhat subjective and lack of dynamical justifications.

My first concern is the choice of the sample. It looks like the energy transport in JJA, according to Fig. 8, is generally associated with the wavenumber 4 to 6. Accordingly, I would expect to find out energy transport to be associated with wavenumber of 4 to 6, regardless of the year. Therefore, the fact that dominant wavenumbers of the energy transport in 2010 is similar to the preferred zonal wavenumber of the quasi-resonant amplification (QRA) theory does not necessarily mean that the energy transport and QRA theory are dynamically connected.

**Reply:** We apologize with the reviewer for the possible misunderstanding: the aim here is not to draw a conceptual link between the QRA mechanism and the existence of poleward meridional energy transports in 2010's JJA. The aim is actually pointing out that:

1. the fact that wavenumbers 4-6 dominate the transports in these events is coherent with Petoukhov-Kornhuber findings about QRA mechanism as an explanation of co-recurrent heat waves in the Northern Hemisphere.
2. the 2010's JJA persistent event is a "typical" extreme event for meridional energy transports in summer, and this motivates looking into the "typicality" of such extreme events, as this opens the possibility to exploit some crucial properties of dynamical chaotic systems, as already pointed out for heat waves and cold spells analysis in the framework of large deviation theory (cfr. Galfi et al. 2019.; Galfi and Lucarini 2021) This is already mentioned at ll. 324-328, but it will be clarified in the revised version of the manuscript;

To this extent, it is not our aim here to establish a dynamical linkage between extreme meridional energy transport events and co-recurrent heat waves in the mid-latitudes. We believe that this would require substantial additional analysis that goes beyond the scope of our work. Rather, we want to emphasize that these extreme events are consistent with a set of typical dynamical patterns, and the recurrence of these patterns deserves further investigation.

**Changes:** we better emphasized the meaning of comparing our results with those explaining co-occurring heatwaves with the QRA mechanism at ll. 391-405..

Also, according to the Figure 11, the extremes are computed with respect to 2010 mean, but shouldn't they be computed with respect to the climatology?

**Reply:** The computation of anomalies has been explored in different ways. In order to account for the trends that are shadowing the patterns of variability, we resolved ourselves to the 2010 seasonal mean. As we agree with the reviewer that this might be a source of confusion, we provided instead a revised version of the composites, with anomalies computed wrt. a detrended seasonal mean over all years.

**Changes:** Anomalies shown in Figure 13 (former Figure 11) are now computed wrt. a detrended seasonal mean over all years.

The second question is the actual dynamical connection between energy transport, QRA mechanism, and heat waves. If I understand correctly, QRA mechanism requires a zonally oriented enhanced jet stream that can act as a strong waveguide. In line with the comments made earlier, with such a zonally oriented background flow, it is little unclear to me how meridional energy transport can be strong. In addition, heat waves are rather caused by processes such as temperature advection, enhanced solar radiation within an anticyclone, and etc. So, if you can discuss how meridional energy transport can dynamically cause (or be associated with) the heat waves, it will help readers to follow the manuscript.

**Reply:** We thank the reviewer for pointing this out. We will add a few sentences suggesting what the possible mechanisms linking QRA and meridional energy transport extremes could be. As mentioned in a previous comment, it is not the main scope of this work drawing such dynamical connection. Despite that, we notice that Petoukhov et al. 2013 already stresses that the quasi-resonant hypothesis is associated with weakened zonal components of the circulation by high-amplitude waves (in their case k=6-8, cfr. their Figure 2), and the authors themselves claim that, "as distinct from Branstator's mechanism, the action of the quasi-resonance mechanism essentially depends on the shape rather than the magnitude of the circumglobal jets." This does not exclude, then, that the QRA mechanism can be associated with extremely strong meridional energy transports. As above, expanding this argument, looking at the dynamical linkages between energy transports, temperature (and moisture) advection in the context of co-recurrent heat waves amplified by QRA mechanism, goes beyond the scope of the work here presented

**Changes:** at ll. 471-482 of the conclusions we stated why we did not enter a detailed discussion of how meridional energy transport extremes relate to the co-occurring heatwaves: this is a matter that shall be treated as the main focus of a successive work, overcoming the zonally integrated approach.

Other Comments:

- L329: It is confusing how composites based on the 30-33 band and 57-60 band can be characterized by negative NAO. The 30-33 band composite is more zonally oriented without a prominent anticyclonic feature over Greenland, and there are almost no signals in the composite by 57-60 band.

**Reply:** We thank the reviewer for pointing out this inconsistency in the interpretation of Figure 12. We actually noticed that there was an error in the panels 30-33 and 45-47 of the equatorward side. We apologize for this mistake. The significance level that we have now introduced helped us to emphasize that the composite maps are the blend of different configurations, of which only the 45-47 features a clear NAO- pattern. Same for the PT pattern, with the North American ridge possibly emerging at all latitudes, but the Pacific trough only significant in the 45-47 band. It was already evident in a previous work (Lembo et al. 2019) that meridional energy transport extremes feature a remarkable latitudinal extension, but often do not extend to the margins of the latitudinal band (cfr. Figure 1f). This is also reflected in former Figure 5a,e. This is a clear example of how, relying on well-established clustering methodology, using (at least in the case of DJF) a well described set of weather regimes, we are able to correctly interpret composite maps of geopotential height anomalies when zonally integrated meridional energy transport extremes occur. This provides a first, though methodologically reasonable, in our opinion, attempt to draw a linkage between the dynamics of the mid-latitudinal eddies and the transport extremes.

**Changes:** we corrected Figure 12, that now contains the correct panels at the right place. The way we interpret composite maps in conjunction with weather regime frequency differences and dominant wavenumbers in the population of extreme events is better described at ll. 354-358 (and at ll. 402-404 and ll. 455-458 for specific cases).

- Decomposing the zonal wavenumber of the energy transport into planetary and synoptic scale is an interesting, and perhaps, an important point, yet their dynamical origin is not discussed well. Therefore, I think the paper can have a broader impact by adding some more discussion on this topic. What are the causes of the planetary vs. synoptic scale meridional energy transports? Is it possible that planetary scale wave and energy transport can be excited by tropical forcing, whereas the synoptic scale waves can be associated with high-frequency transient eddy fluxes? If one can speculate the cause of those energy transport at different zonal scales, it might be beneficial to diagnose the variability and intraseasonal fluctuations of meridional energy transport and perhaps the long-term changes under anthropogenic warming. I will let the authors to decide whether to add a discussion on this topic.

**Reply:** We thank the reviewer for the useful suggestion. In a previous paper (Lembo et al. 2019), the authors argued that the planetary-scale meridional energy transports were of very different nature in the two hemispheres. In particular, we found that planetary-scale waves in the Southern Hemisphere were well present in the wintry season, and they could only be explained in terms of transient waves; this could be seen only via wavenumber decomposition, whereas the classic stationary-transient decomposition was not capable of doing that, with stationary transport almost negligible in the Southern Hemisphere (cfr. Dell'Aquila et al. 2007). Recently, Shaw 2014 found that planetary waves have a peculiar role in the abrupt seasonal transition of the Northern Hemisphere, with a SST threshold governing the latent and momentum transport, thus potentially involving the seasonal development of monsoons. This suggests that an in-depth analysis of the dynamical mechanisms leading to the development of planetary waves, and its interaction with synoptic-scale waves, would involve breaking down the moist static energy into its components (dry static energy component and

latent energy), an explicit consideration of moisture transport, and of course a proper selection of the wavenumber in terms of the wavelength that is actually representing the process. Parts of these discussions have been carried out in a paper recently under review on WCD, by Stoll and Graversen. This could be a basis to move forward towards an analysis of regional extreme transports, in connection with geopotential, pressure and SST patterns.

**Changes:** the investigation of how counter-gradient planetary scale components in the summer equatorward extremes are related to the circulation, especially in the Southern Hemisphere, will be the focus of a future work.

Minor comment

- L179: Here, the patterns are based on the time period of 1979-2013, while L62 says that the analyzed time period is 1979-2012. If this is not a typo, then I think it is better to use the same time period for all analysis.

**Reply:** This is a typo which we will correct. The considered period is 01/01/1979-31/12/2012.

**Changes:** l. 216 has been corrected accordingly.

- L207, L337, and Figure 11 caption: It is better to spell out 'with respect to' instead of just writing wrt.

**Reply:** Thank you for noticing it, we will expand the acronym in the revised manuscript.

**Changes:** wrt. acronym has been expanded to "with respect to" at every occurrence.

- L220 and L222: I think it is better to indicate specific latitudinal band instead of expressing as 'edges of the mid-latitudinal channel' or 'high/low latitude'.

**Reply:** As already mentioned above, we provided a new figure with skewness as a function of the latitudinal bands in the revised manuscript.

**Changes:** this part has been completely rearranged.

- For clarity, it would be good to clearly indicate which figures authors are referring to. For example, L252, "… frequency of NAO-(Fig5a), AO(Fig.5e), and PT (Fig. 5c)" and L253 "In JJA, NHC4(Fig. 6e)/EATC2(Fig.6a) …". Same clarification in other lines will help readers to follow the manuscript better.

**Reply:** Thanks. We have proceeded as suggested by the reviewer.

**Changes:** ll. 314-331 have been modified accordingly.

- It is somewhat difficult to remember the physical pattern of all the JJA pattern with the current names (for example, L276 and 278 EATC2 and NHC4 / EATC4, PACC4, NHC3). So, I suggest to re-name JJA patterns with more intuitive or commonly known names as in DJF, or explicitly explain in the text. For example, L276 can be re-written as '…EATC2 and NHC4(Scandinavia blocking-like pattern) …'.

**Reply:** We were actually thinking of a possible better way to rename the weather regimes in summer. Unfortunately, literature does not help as it did for winter. We believe, though, that any naming based on recognized patterns would be subjective. This is why we preferred staying with the original formalism.

**Changes:** the choice of the acronyms for JJA weather regimes have been left unchanged.

- L381-398: In these paragraphs, references are written without parenthesis. For example, L383 should be written as "… atmospheric features (Galfi et al. (2019); … et al. (2021))".

**Reply:** Thanks. We implemented this in the revised manuscript.

**Changes:** All missing parentheses have been added to the references.

Figures

- Figures 2 a-d / A1 / A2: I think it might be visually beneficial to use rectangular map instead of circular map if you wish to only plot certain designated domain. This is only a suggestion, so I will let the authors to decide.

**Reply:** We made several attempts on how to best visualize the weather regime patterns. We also tested a rectilinear grid, but we noticed that it was overrepresenting the high latitudes, and it was not very clear what the pattern would be at the mid-latitudes. Further, as we needed to plot 4*3*2 maps, rectangular plots would have forced us to spread the information on several figures. Also, it would not have been immediately clear what is the regional domain where the k-mean clustering would be performed. For these reasons, we finally opted for the polar projection.

**Changes:** Figure 2 has been changed in order to minimize blank spaces.

- Figures 3b-e and 4b-c: having a same x-axis range for all four panels will make it easier to compare the relative magnitude of the transport for different wave number regimes. Also x-label should be 'Meridional energy transport', not heat transport.

**Reply:** Thanks. We agree with the reviewer that having the same x-axis range would ease comparison of relative magnitudes. At the same time, though, it would make it more difficult to appreciate the features of the extreme PDFs, where the magnitude is smaller, e.g. Figure 3d. For this reason, we would retain the chosen x-axis range and add, possibly as an inset, or as a different figure, a plot of the relative magnitude of the four wave groups as a function of latitude.

**Changes:** we proposed in the new Figures 3 and 4 a season-wise homogeneous choice of the x-axis range, so that the components can be compared within the same season, but not across the seasons.

- Figure 9 and 10: (a) and (b) are missing. Also, the unit of color bars in Fig. 9, 10, and 11a are [Pa], which is not [dam].

**Reply:** Thanks. We corrected that in the revised manuscript.

**Changes:** Figures 11, 12 and 13 now contain the right labels and the right unit of measure for z500.

---

## Author Response (AR2)

**Reviewer 2**

The authors have put many efforts to improve the quality and the clarity of the paper. For example, authors expanded the Introduction and included additional discussion on their methodologies, such as choice of the wavenumber threshold, as an appendix. In addition, I found that more detailed scientific and meteorological interpretations have been added to the text, which definitely helps reader to understand the result in depth. For future review process, though, it will be helpful if authors can provide additional notes on the line numbers where changes have been made. It was somewhat difficult to follow which portion of the main text was updated. I recommend this paper to be accepted, but please correct the typo (see below).

**Reply:** we thank the reviewer for appreciating our effort to improve and clarifying the manuscript. We apologise for not having sufficiently pointed towards the lines in the new version of the manuscript where changes have been performed. These were mentioned in the section "Changes" of the reply to reviewer document, but we could have missed referring to them in some of the comments.

Minor comment
1. Typo: I think the figure caption of Figure 4 should be "same as in Figure 3…".

**Reply:** thanks for pointing this out. The caption has been corrected accordingly.